# Reversible glycosidic switch for secure delivery of molecular nanocargos

Pierre-Alain Burnouf[1,2,3], Yu-Lin Leu[4], Yu-Cheng Su[2], Kenneth Wu[2], Wei-Chi Lin[4] & Steve R. Roffler[2,5]

Therapeutic drugs can leak from nanocarriers before reaching their cellular targets. Here we describe the concept of a chemical switch which responds to environmental conditions to alternate between a lipid-soluble state for efficient cargo loading and a water-soluble state for stable retention of cargos inside liposomes. A cue-responsive trigger allows release of the molecular cargo at specific cellular sites. We demonstrate the utility of a specific glycosidic switch for encapsulation of potent anticancer drugs and fluorescent compounds. Stable retention of drugs in liposomes allowed generation of high tumor/blood ratios of parental drug in tumors after enzymatic hydrolysis of the glycosidic switch in the lysosomes of cancer cells. Glycosidic switch liposomes could cure mice bearing human breast cancer tumors without significant weight loss. The chemical switch represents a general method to load and retain cargos inside liposomes, thereby offering new perspectives in engineering safe and effective liposomes for therapy and imaging.

[1] Taiwan International Graduate Program in Molecular Medicine, National Yang-Ming University and Academia Sinica, Taipei 11529, Taiwan. [2] Institute of Biomedical Sciences, Academia Sinica, Taipei 11529, Taiwan. [3] Institute of Biochemistry and Molecular Biology, National Yang-Ming University, Taipei 11221, Taiwan. [4] Department of Pharmacy, Chia Nan University of Pharmacy and Science, Tainan 71710, Taiwan. [5] Graduate Institute of Medicine, College of Medicine, Kaohsiung Medical University, Kaohsiung 80708, Taiwan. Correspondence and requests for materials should be addressed to S.R.R. (email: sroff@ibms.sinica.edu.tw)

Nanomedicine is a promising approach to increase the selectivity and efficacy of therapeutic agents. Encapsulation of therapeutic agents in nano-carriers can facilitate selective targeting of inflammatory diseases and tumors due to the enhanced permeability and retention effect (EPR), a phenomenon in which nanomedicines can diffuse through relatively large pores in blood vessels and accumulate at disease sites[1]. Selective accumulation of nanomedicines in diseased tissues by the EPR effect relies on passive convection and diffusion over a period of days[2–4]. Many important therapeutic agents, however, are rapidly released from nanocarriers under physiological conditions, leading to systemic side effects[3,5]. Importantly, leakage of drugs from nanocarriers during the long circulation times required for adequate uptake of nanomedicines into poorly perfused tissues and tumors can result in delivery of suboptimal drug concentrations[6].

Liposomes, made of a lipid bilayer with an internal aqueous core, represent the preponderance of nanoparticles approved by the Food and Drug Administration (FDA) or in clinical phase II/III trials[7,8]. The popularity of liposomes reflects their biocompatibility, low immunogenicity, and prolonged half-life in the circulation[9,10]. Despite the tremendous success of liposomes in

drug delivery[11,12], poor retention of important and potent therapeutic agents remains a major roadblock in the field of nanomedicine[13,14]. This is important because stably retained drugs allow greater tumor accumulation than unstable drugs (>1% vs. 0.1%) and consequently display stronger anticancer activity and overall survival in vivo[15]. Elegant strategies have been developed to load and retain drugs in liposomes[16–21], but these approaches generally pertain to specific drug molecules. For example, docetaxel was modified by a N-methylpiperazinyl butanoic acid group to produce a protonated derivative to allow the use of a pH gradient for improved loading[20]. In another approach, docetaxel was modified by attachment of a glucose group to increase water solubility and improve loading capacity in liposomes by avoiding membrane accumulation of docetaxel[21]. Although glycosylated docetaxel retained 90.9% of tubulin stabilization, such modification can strongly affect drug potency in other cases. Here we propose a general chemical approach to actively load and stably retain cargos in liposomal carriers. A chemical switch is covalently attached to a cargo molecule via a cue-responsive trigger (Fig. 1a). The switch can be interconverted between two states by defined environmental conditions (i.e., pH,

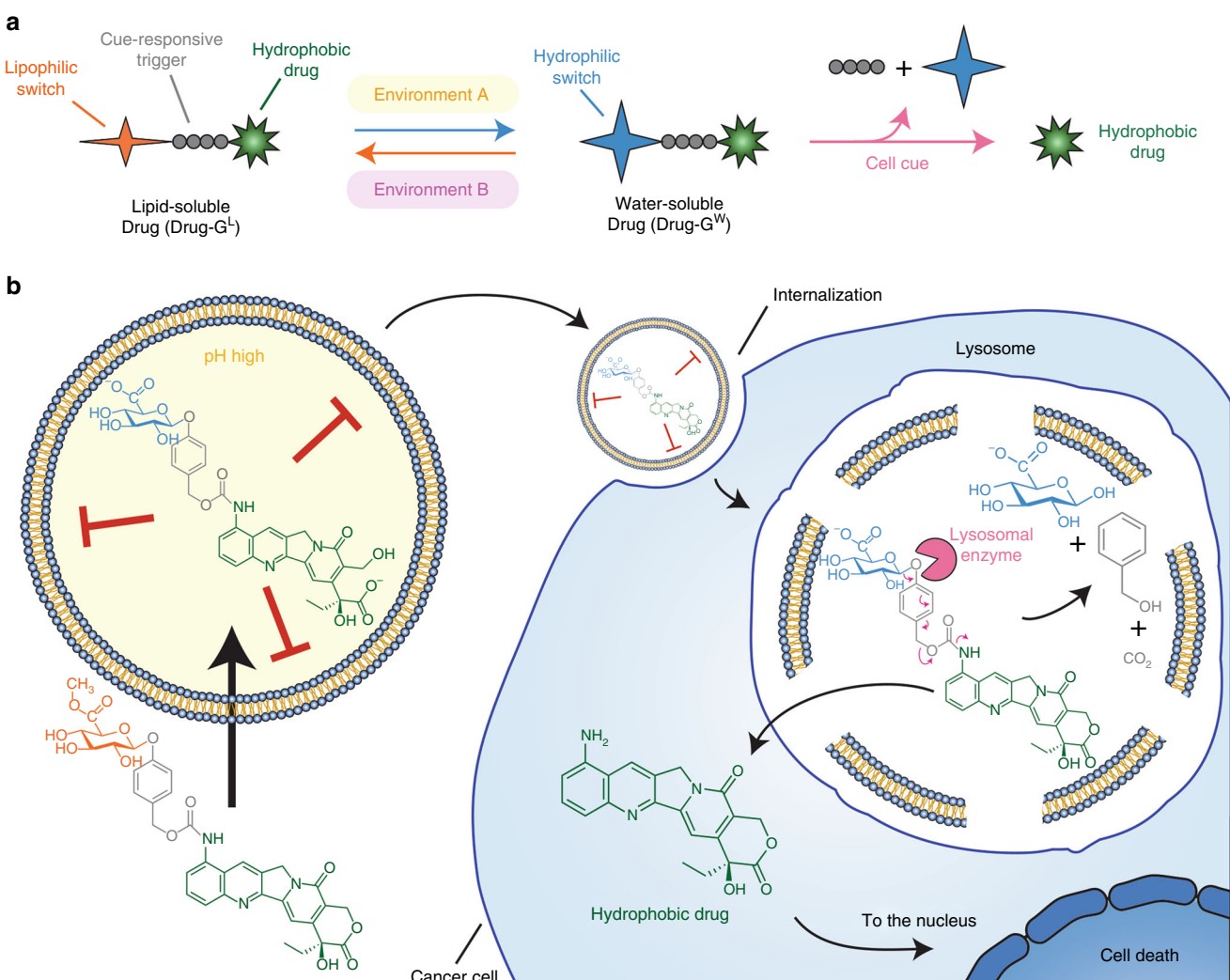

**Fig. 1** Concept of the reversible glycosidic switch. **a** A water/lipid soluble switch is conjugated to a hydrophobic cargo via a cue-responsive linker able to release the cargo at a specific site. **b** We employed a glycosidic switch that can be esterified in acidic methanol to a lipid-soluble form (shown in orange) to facilitate active loading across the liposome membrane. At a high internal pH in the aqueous lumen of the liposomes, the glycosidic switch is spontaneously saponified to the water-soluble form (shown in blue), resulting in strong retention of the molecular cargo. The cargo loaded in the liposomes is stable in the circulation and can be delivered to target cells for cellular uptake and lysosomal degradation. The cue-responsive trigger (shown in grey) responds to beta-glucuronidase (shown in pink) in lysosomes to release the parental cargo within the cells

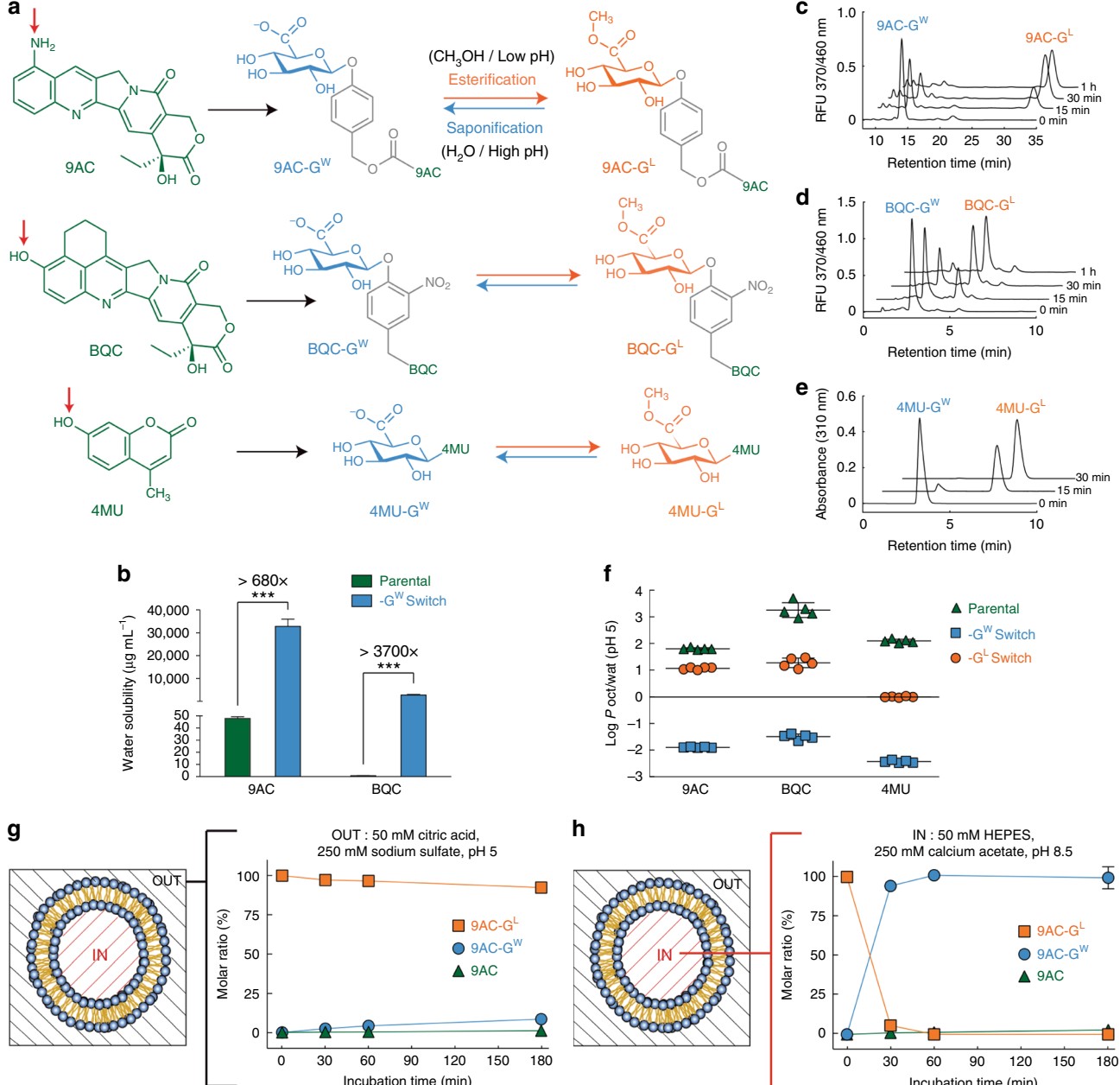

**Fig. 2** Effect of switch conjugation on hydrophobic cargos. **a** The switch was conjugated to 9-aminocamptothecin (9AC) and 5,6-dihydro-4H-benzo[de] quinoline-camptothecin (BQC) via a cue-responsive trigger[17,24] to generate 9AC-G$^W$ and BQC-G$^W$, while it was directly attached to 4-methylumbellirone to generate 4MU-G$^W$ (the red arrows indicate the position where the glycosidic switch was conjugated). The carboxylic acid function of the switch is converted in acidic methanol to a lipid-soluble methylester, which can be reconverted to the water-soluble form in basic aqueous solutions. **b** Comparison of the water solubility of the parental drugs 9AC and BQC with their water-soluble switch conjugated forms (9AC-G$^W$ and BQC-G$^W$). Error bars, SD, $n = 3$. **c** Kinetics of the conversion of 9AC-G$^W$ to lipid-soluble 9AC-G$^L$, **d** BQC-G$^W$ to BQC-G$^L$ and **e** 4MU-G$^W$ to 4MU-G$^L$ in acidic methanol at 62 °C. The HPLC chromatograms show samples at 0, 15, 30, and 60 min. RFU relative fluorescent units. **f** Partition coefficients (Log scale) between 50 mM citric acid buffer pH 5 and 1-octanol of the parental compounds compared to their respective -G$^W$ and -G$^L$ forms. Error bars: SD, $n = 5$. **g** Stability of 9AC-G$^L$ in liposomal external buffer (50 mM citric acid, 250 mM sodium sulfate, pH 5) at 75 °C. Error bars: SD, $n = 3$. **h** Conversion of 9AC-G$^L$ to 9AC-G$^W$ in liposomal internal buffer (50 mM HEPES, 250 mM calcium acetate, pH 8.5) at 75 °C. Error bars: SD, $n = 3$. Statistical significance of differences in mean values: ***$p < 0.0001$

redox, ion concentration) present outside or inside the liposomes, respectively. In the lipophilic state, the switch promotes passage of the cargo across the liposomal membrane whereas in the hydrophilic state, the switch helps retain the cargo inside the aqueous core of the liposome (Fig. 1b). Upon contact with an appropriate cell location, the cue-responsive trigger is activated to release the original cargo molecule. The method may be a more general approach as we demonstrated the utility of the switch

concept by stably retaining chemically different hydrophobic drugs in liposomes.

## Results and Discussion

**Glycosidic switch drug conjugates.** To load and retain hydrophobic cargos in liposomes, a glycosidic switch that can alternate between hydrophilic and hydrophobic states was attached to

target molecules via a cue-responsive trigger. The switch is (1) lipid-soluble (drug-$G^L$) at acidic pH for active loading into liposomes, (2) water-soluble (drug-$G^W$) at elevated pH for stable retention inside liposomes, and (3) releases the original molecular cargo upon enzymatic hydrolysis of the trigger inside target cells (Fig. 1b). We wish to emphasize that the term "-$G^W$" used with camptothecin drugs refers either to the closed lactone or the opened carboxyl forms, which depends on the environmental pH. As an example, "9AC-$G^W$" inside the liposomes, refers to the open carboxyl form, however, when released inside the lysosomes "9AC-$G^W$" is expected to reform the closed lactone ring form. To investigate the glycosidic switch for liposomal loading, we selected two hydrophobic anticancer drugs, 9-aminocamptothecin (9AC) and 5,6-dihydro-4H-benzo[de]quinoline-camptothecin (BQC) and one fluorescent compound 4-methylumbelliferone (4MU) that also displayed anticancer activity in recent reports[22,23]. The drugs, 9AC and BQC, are characterized by very low water solubility (<50 μg mL$^{-1}$) and potent anticancer activity against a wide range of human cancer cells[24,25]. 4MU was chosen as a representative of non-camptothecin compounds to demonstrate the utility of the glycosidic switch in a more general approach. The switch was either linked to 9AC and BQC via cue-responsive triggers to produce 9AC-$G^W$ and BQC-G$^{[W26}$ or directly attached in the case of 4MU-$G^W$ (Fig. 2a). The water solubility of 9AC-$G^W$ and BQC-$G^W$ were 680 and 3700 fold greater than the parental hydrophobic drugs 9AC and BQC, respectively (Fig. 2b).

More lipophilic cargos (9AC-$G^L$, BQC-$G^L$, and 4MU-$G^L$) were generated by esterification of the switch in acidic methanol at 65 °C. HPLC analysis showed that high levels of 9AC-$G^L$ (Fig. 2c), BQC-$G^L$ (Fig. 2d), and 4MU-$G^L$ (Fig. 2e) formed within 1 h. Measurement of the partitioning of the compounds in an octanol-

water two-phase system demonstrated that the parental compounds were hydrophobic, whereas 9AC-$G^L$, BQC-$G^L$, 4MU-$G^L$ possessed slightly lipophilic behavior, able to dissolve in water and octanol but with a preference for the organic phase for 9AC-$G^L$ and BQC-$G^L$ (Fig. 2f). A lipophilic behavior is beneficial for facilitated loading of drugs in liposomes because it allows passage through the liposomal membrane and accumulation in the aqueous lumen. By contrast, 9AC-$G^W$, BQC-$G^W$, and 4MU-$G^W$ partitioned in the aqueous phase, suggesting that they may remain inside the lumen of liposomes where the bilayer would act as a physicochemical barrier.

The lipid-soluble glycosidic switch could respond to environmental conditions; 9AC-$G^L$ was stable at 75 °C at pH 5 (Fig. 2g), but was rapidly converted to the highly water-soluble 9AC-$G^W$ at pH 8.5 (Fig. 2h). We expect that 9AC-$G^L$, BQC-$G^L$, and 4MU-$G^L$ in an external low pH buffer should pass through the liposomal membrane for efficient active loading, while a high pH in the liposome lumen should allow conversion and accumulation of the hydrophilic 9AC-$G^W$, BQC-$G^W$, and 4MU-$G^W$. It is worth noting that camptothecin drugs (such as 9AC and BQC) might be well suited for improved retention due to the presence of a lactone ring. Indeed, the lactone ring will be found under an open form at the high pH inside the liposomes, revealing an additional charged carboxylate group for enhanced retention. After lysosomal routing and enzymatic degradation of the liposomes[27], the closed lactone form can spontaneously reform at the acidic pH in lysosomes for rapid escape into the cytosol and enhanced anticancer activity afforded by the lactone form of camptothecins[28].

**Active drug loading to liposomes**. Liposomes composed of 65% 1,2-distearoyl-sn-glycero-3-phosphocholine (DSPC), 30%

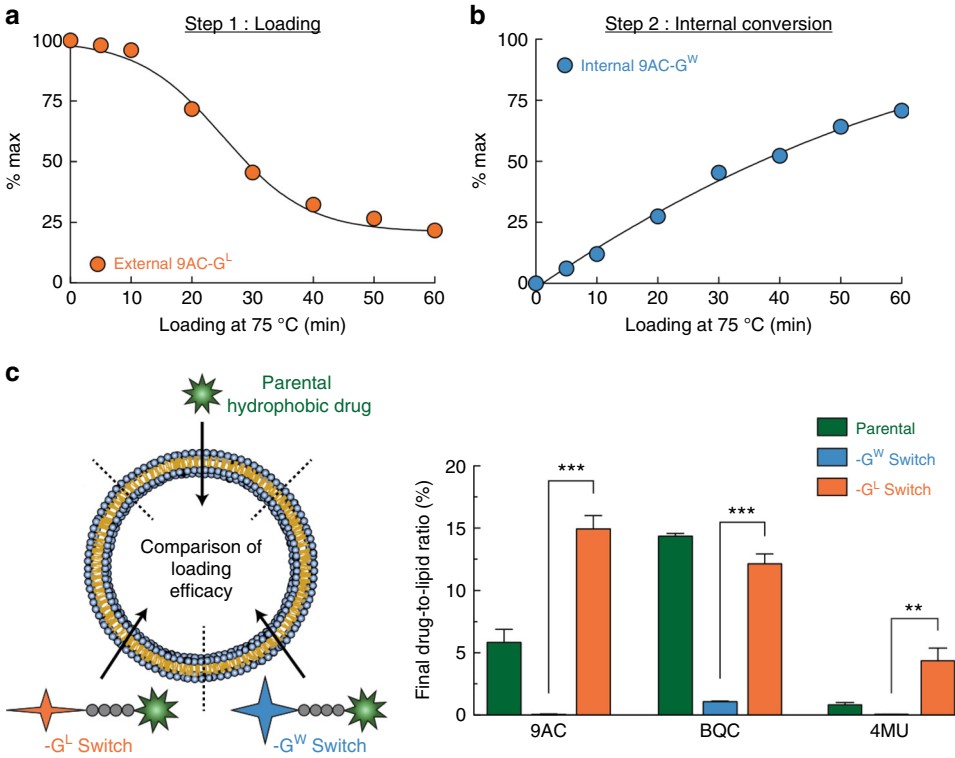

**Fig. 3** Liposomal encapsulation of glycosidic switch cargos. **a** Kinetics of the disappearance of 9AC-$G^L$ at 75 °C from the external low pH medium into liposomes with an internal high pH gradient. **b** Kinetics of the generation of 9AC-$G^W$ inside liposomes. **c** Comparison of the final drug-to-lipid molar ratios for parental drugs vs. the glycosidic switch forms of the parental compounds in liposomes. Error bars: SD, $n = 3$. Statistical significance of differences in mean values: **$p < 0.001$ and ***$p < 0.0001$

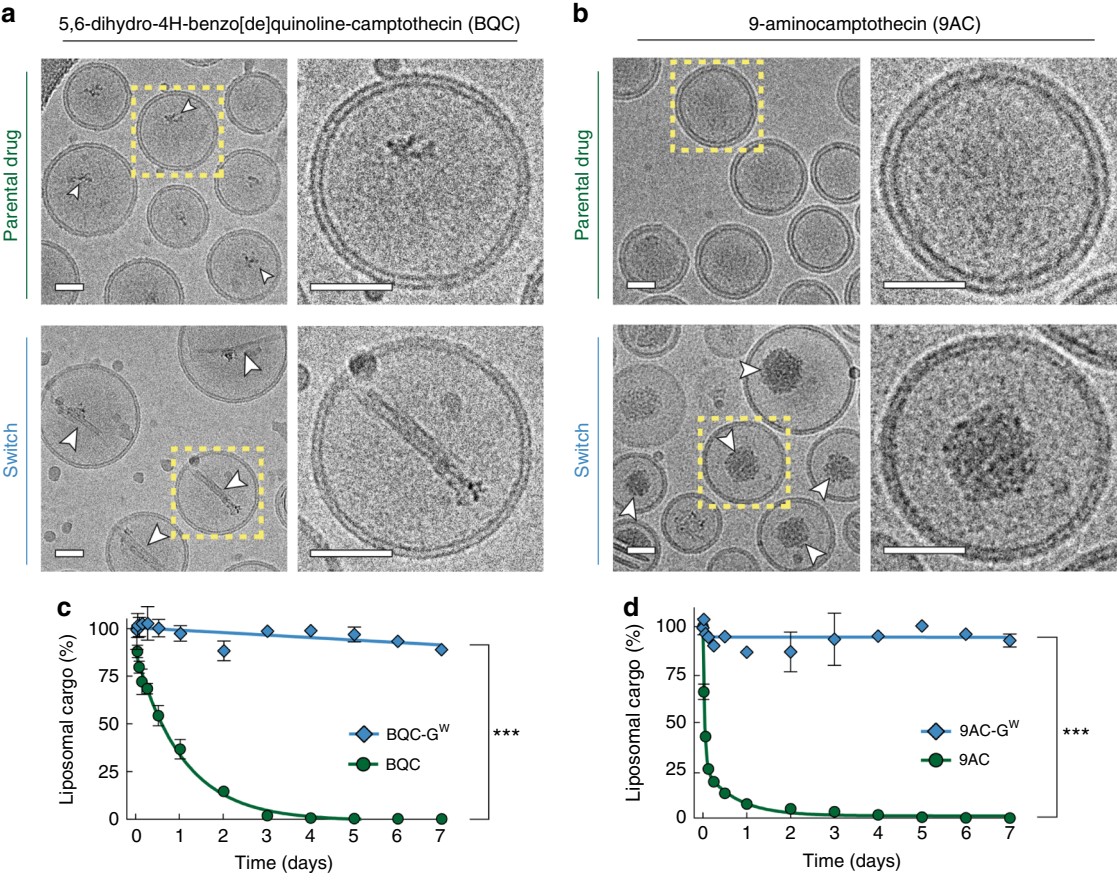

**Fig. 4** Imaging and retention of 9AC-G$^W$ and BQC-G$^W$ in liposomes. **a** Cryogenic electron micrograph of liposomes loaded with BQC (upper panel) and BQC-G$^W$ (lower panel). **b** Cryogenic electron micrograph of liposomes loaded with 9AC (upper panel) and 9AC-G$^W$ (lower panel). Arrowheads show internal precipitated drug, and yellow dashed squares represent the selected liposomes for enlargement. Scale bar = 35 nm. **c** Drug release kinetics of BQC as compared to BQC-G$^W$ and **d** 9AC as compared to 9AC-G$^W$. Error bars: SD, $n = 3$. Statistical significance of differences in mean values: ***$p < 0.0001$

cholesterol and 5% 1,2-distearoyl-sn-glycero-3-phosphoethano-lamine-N-[methoxy(PEG)-2000] were prepared containing an internal buffer of 50 mM HEPES and 250 mM calcium acetate at pH 8.5. The external medium of the liposomes was exchanged by size exclusion chromatography to 50 mM citric acid, 250 mM sodium sulfate pH 5. The ready-to-load liposomes with an internal pH 8.5 and external pH 5 showed a uniform unilamellar distribution (Supplementary Fig. 1a) and an average size of 135 nm, as determined by dynamic light scattering (Supplementary Fig. 1b).

Incubation of 9AC-G$^L$ with liposomes at a molar ratio of 1:4.6 (drug:lipid) at 75 °C revealed disappearance of 9AC-G$^L$ in the external medium (Fig. 3a) and corresponding accumulation of water-soluble 9AC-G$^W$ inside liposomes (Fig. 3b), consistent with the conversion of 9AC-G$^L$ to 9AC-G$^W$ by saponification at the basic pH inside the liposomes. As the liposomal fraction was separated from the external medium and lysed with detergent to reveal its contents, this provides direct evidence that the ester group of the lipophilic switch was hydrolyzed inside the liposomes. 9AC-G$^L$ and BQC-G$^L$ were loaded with efficiencies of 71.3% and 56.2% respectively (Supplementary Fig. 2a), and were internally converted to the water-soluble (-G$^W$) forms with efficiencies of 96.6 and 99.6% after 1 h (Supplementary Fig. 2b). The final drug-to-lipid molar ratios were 14.9 and 12.1% (Supplementary Fig. 2c). After loading, the glycosidic switch is exposed to an internally high pH and undergo saponification, releasing methanol. As methanol diffuses through lipid bilayers[29],

it is not retained internally and is diluted in a larger external volume. Removal of non-encapsulated compounds after loading also contributes to remove the methanol generated during that step. We also compared loading of the parental hydrophobic drugs and water-soluble switch compounds in the same liposomal formulation. 9AC-G$^L$ loaded about three fold more efficiently than 9AC and 4MU-G$^L$ loaded about five fold more efficiently than 4MU whereas BQC-G$^L$ and BQC displayed similar loading efficiencies (Fig. 3c). This result suggests that the glycosidic switch can be employed for liposomal loading of a wide range of compounds. By contrast, 9AC-G$^W$, BQC-G$^W$, and 4MU-G$^W$ could not be effectively loaded into liposomes, likely due to inability to cross the lipid bilayer. As DMSO is known to enhance the permeability of lipid bilayers, we used the same DMSO concentration (9%, v:v) for all the loading conditions to ensure that any improvement in drug loading efficiency is solely attributed to the influence of the glycosidic switch. 4MU-G$^L$ was loaded less efficiently than 9AC-G$^L$ and BQC-G$^L$. We suspect that the partitioning of the modified drugs between the aqueous and the organic phases is important for efficient loading. For example, 4MU-G$^L$ is only very slightly lipophilic (Log $P \sim$ −0.002) and it appears that a clear partitioning towards the organic phase is preferred, as observed with BQC-G$^L$ (Log $P \sim$ 1.3) and 9AC-G$^L$ (Log $P \sim$ 1.1) for efficient loading. We suspect that esterification of the glycosidic switch with longer carbon chains (i.e., ethanol, propanol, etc.) might help to increase the log $P$ values of such compounds and therefore the loading efficiency.

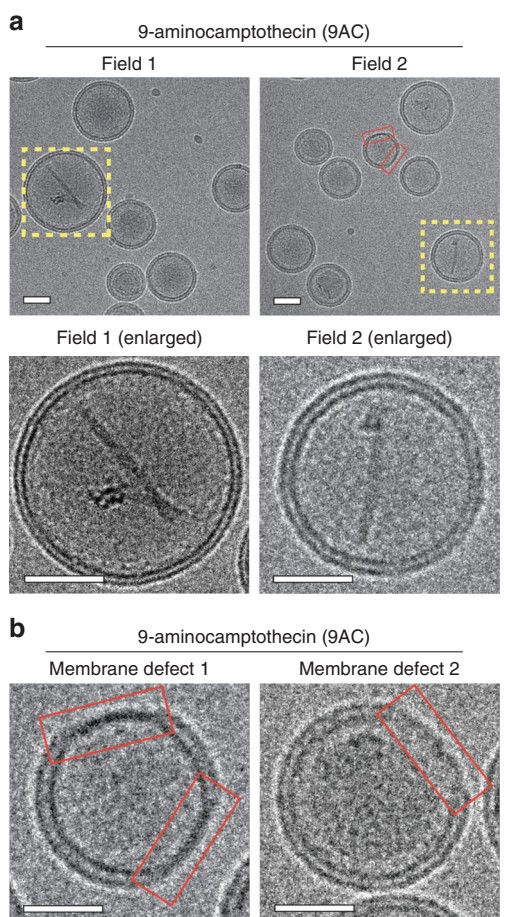

**Fig. 5** Visualization of membrane defects in 9AC liposomes. **a** Liposomes were loaded with 9AC and imaged by cryogenic electron microscopy. Only a small fraction of liposomes contained precipitated 9AC in the core (yellow dashed squares, upper panel) more clearly visible at an enlarged scale (lower panel). Scale bar = 35 nm. **b** Several liposomes presented defects in the membrane bilayer in which the bilayer appears to collapse in a thin monolayer (red boxes) after loading with 9AC. Two representative examples are shown. Scale bar = 25 nm

In the end, the loading efficiency of 4MU-$G^L$ was still superior to the parental compound 4MU (Fig. 3c).

**In vitro release kinetics**. A key goal of the glycosidic switch is to stably retain cargos within the lumen of liposomes. Cryogenic electron microscopy imaging revealed a precipitate in the lumen of liposomes loaded with BQC-$G^W$ (Fig. 4a) and 9AC-$G^W$ (Fig. 4b). Some drug precipitate was also present in the lumen of liposomes loaded with BQC, although to a lesser degree as compared with BQC-$G^W$. The difference in terms of core precipitation for 9AC and 9AC-$G^W$ was even more obvious (Fig. 4b). Some liposomes loaded with 9AC also contained precipitate (Fig. 5a), but the majority of liposomes appeared clear. The calcium ions in the liposomes likely form a complex with the glycosidic switch under a water-soluble form (-$G^W$), by reaction between positively charged calcium and negatively charged carboxylate of the switch, leading to precipitation inside the liposomes, which may help retain molecular cargos during delivery[30]. We compared the retention of parental drugs or the corresponding glycosidic switch drugs in a dialysis-based assay in PBS supplemented with 10% fetal bovine serum (FBS) at 37 °C over a period of seven days (Supplementary Fig. 3). Encapsulated BQC

and 9AC rapidly diffused out of liposomes with loss of about 60% and 90% within 1 day, respectively (Fig. 4c, d). We observed a significant increase in the membrane thickness of liposomes loaded with 9AC and BQC as compared to empty liposomes (Fig. 6a, b), consistent with substantial partition of the drugs into the lipid bilayer, leading to rapid diffusion into biological fluids[31,32]. Such nano-carriers may act more as sustained-release vehicles rather than true nanomedicines that rely on the EPR effect to achieve selective tumor targeting, which requires days for maximal nanocarrier accumulation[33,34]. Possible defects in the membrane bilayer of 9AC liposomes were also noticed (Fig. 5b), which may be due to destabilization of the membrane by interaction of 9AC and membrane phospholipids[35–38]. Liposomal BQC-$G^W$ and 9AC-$G^W$, on the other hand, displayed high stability with 93% and 90% of drug retained inside liposomes after a period of 7 days, respectively. A beneficial effect of the camptothecin lactone ring on improved retention can also be considered, as the lactone ring is present in the open charged carboxyl form at the high pH inside liposomes. However, the poor retention of 9AC and BQC in the same liposomes (with high internal pH) demonstrates that opening of the lactone ring by itself is insufficient to achieve good retention of the drugs inside liposomes. BQC-$G^W$ and 9AC-$G^W$ liposomes also exhibited reduced membrane thickness as compared to 9AC or BQC liposomes (Fig. 6a, b), implying less accumulation of the drugs in the lipid bilayer. The improved retention of glycosidic switch compounds in the lumen of liposomes may provide sufficient time for target-selective delivery of nanomedicines. The glycosidic switch also reduces drug potency (Supplementary Fig. 4)[24,25], thereby affording an additional level of safety.

**The glycosidic switch is reversible**. The cue-responsive trigger was designed to release molecular cargos upon reaction with lysosomal beta-glucuronidase (Fig. 7a), which may be elevated in cancer cells[39]. The reversibility of the switch was first examined in vitro by HPLC analysis of (1) 9AC, (2) 9AC-$G^W$ formed by conjugation of the glycosidic switch to 9AC, (3) 9AC-$G^L$ generated by incubation of 9AC-$G^W$ in acidic methanol, (4) 9AC-$G^W$ regenerated from 9AC-$G^L$ by alteration of the pH to basic conditions (using the pH 8.5 liposomal internal solution used during loading), and (5) 9AC reformed by addition of purified human beta-glucuronidase in a pH 4.5 buffer at 37 °C (to mimic lysosomal conditions) for 30 min to 9AC-$G^W$ (Fig. 7b). These results show that the glycosidic switch can be interconverted between lipophilic and hydrophilic forms as well as be triggered by lysosomal beta-glucuronidase to release parental drug.

To examine if the cue-responsive trigger functioned in situ, we added 9AC-$G^W$ liposomes to HCC36 human liver cancer cells or MDA-MB-468 human breast cancer cells and monitored the appearance of 9AC in the culture medium by HPLC (Fig. 7c). We seeded cells at high density to limit cell mitosis and minimize cell death because camptothecin drugs are selectively toxic during the S-phase of the cell cycle. 9AC progressively accumulated in the culture medium of both HCC36 and MDA-MB468 cells, but not in wells in which no cells were seeded (Fig. 7d), indicating that liposomal 9AC-$G^W$ could be converted to 9AC in cancer cells. The amounts of 9AC accumulating in the medium of HCC36/ anti-PEG cells, which express chimeric receptor that accelerates the endocytosis of PEGylated compounds[40], was about five-fold greater than in HCC36 cells after 72 h of incubation with 9AC-$G^W$ liposomes (Fig. 7e), consistent with the need to internalize liposomes for activation of the cue-responsive trigger. Addition of a beta-glucuronidase inhibitor (D-saccharic acid 1,4-lactone) or RNAi to knock down beta-glucuronidase in the lysosomes of HCC36/anti-PEG cells significantly reduced the generation of

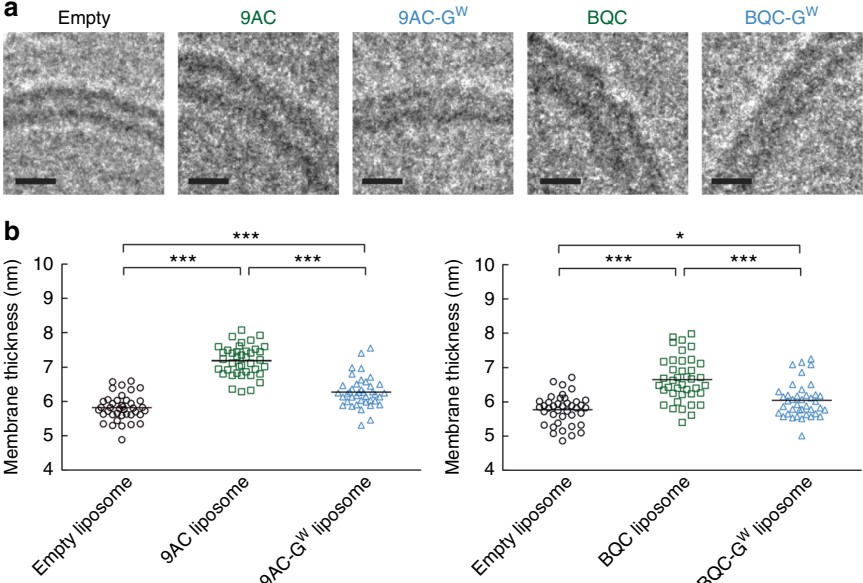

**Fig. 6** Membranes thickness of drug-loaded liposomes. **a** Cryogenic electron micrographs of the membranes of liposomes were measure before and after loading with 9AC, 9AC-G$^W$, BQC, and BQC-G$^W$. Scale bar = 8 nm. **b** Membrane thickness for the different liposomal formulations. Values were determined using ImageJ. Stars indicate significant differences between mean values: *$p < 0.05$ and ***$p < 0.0001$. $n = 40$

9AC as compared to HCC36/anti-PEG cells, highlighting the importance of beta-glucuronidase in trigger activation.

Functioning of the cue-responsive trigger was also visualized in HCC36/anti-PEG cells incubated with fluorescein-G$^W$ liposomes, which are non-fluorescent until beta-glucuronidase cleaves the switch to release fluorescein (Fig. 8a). DiD-labeled fluorescein-G$^W$ liposomes displayed cellular uptake of the liposomes (red) and lysosomal (magenta) regeneration of fluorescein (green), which then diffused into the cell cytoplasm, demonstrating triggered removal of the glycosidic switch from fluorescein-G$^W$ inside the cells (Fig. 8b). Taken together, these results show that the cue-responsive trigger allows generation of the original molecular cargo after internalization of liposomes and reaction with lysosomal beta-glucuronidase. Camptothecin-based cargos may be particularly suitable for lysosomal-responsive triggers since the active lactone form will be generated at the acidic pH in lysosomes[41].

**In vitro cytotoxicity of 9AC-G$^W$ liposomes.** We assessed the cytotoxicity of liposomal 9AC-G$^W$ against four human colon cancer cells (LS174T, HM7, HT29, and HCT116), three human lung cancer cells (CL1-5, NCI-H2170, and SK-MES-1) and one human breast cancer cell line (MDA-MB-468). We chose clinically-approved PEGylated liposomal doxorubicin (Doxisome) for comparison to examine the activity of 9AC-G$^W$ liposomes relative to an established stable liposomal drug formulation that is used clinically for the treatment of patients suffering from ovarian cancer, AIDS-related Kaposi's sarcoma and multiple myeloma. Liposomal 9AC-G$^W$ displayed significantly greater anti-proliferative activity than Doxisome against LS174T, HM7, HT29, HCT116, CL1-5, NCI-H2170, and MDA-MB-468 cells and equivalent toxicity against SK-MES-1 cells (Supplementary Fig. 5a, b & Supplementary Table 1). We also examined the anti-proliferative effect of 9AC-G$^W$ liposomes to non-cancerous human fibroblasts (GM637), murine liver cells (BNL CL.2), and murine fibroblasts (3T3) (Supplementary Fig. 6). We noted that the IC$_{50}$ values of human fibroblasts treated with 9AC-G$^W$ liposomes was higher than any of the human cancer cells tested

(1.6 μM vs. 0.1 ~ 1.2 μM) while murine fibroblasts and liver cells were even less sensitive to 9AC-G$^W$ liposomes with IC$_{50}$ values of 16 and 22 μM, respectively. We believe that 9AC-G$^W$ liposomes demonstrate reduced toxicity to normal cells due to two main reasons: (1) The doubling time of normal cells is usually slower than cancerous cells, and (2) The overexpression of lysosomal enzymes such as beta-glucuronidase by cancerous cells[39]. Several byproducts of the degradation of the glycosidic switch, including benzyl alcohol and glucuronate, are produced during liposomal processing within the cells. However, their toxicity is several orders of magnitude below the toxicity of camptothecins. We conclude that 9AC-G$^W$ liposomes display at least as much in vitro anticancer activity as Doxisome.

**In vivo evaluation of 9AC-G$^W$ liposomes.** To investigate if stable retention of drugs in liposomes afforded by the glycosidic switch translates into enhanced circulation half-life in vivo, we measured the concentration of 9AC-G$^W$ in serum samples after intravenous delivery of 35 μmol kg$^{-1}$ 9AC-G$^W$ liposomes or the same concentration of free 9AC-G$^W$ to 12 to 16 weeks-old female mice. Figure 9a shows that free 9AC-G$^W$ was rapidly cleared from the circulation of mice (initial half-life = 9.3 min; terminal half-life = 38.6 min) whereas liposomal 9AC-G$^W$ displayed much slower clearance (initial half-life = 36.5 min; terminal half-life = 10.1 h). At 6 h after injection, the concentration of liposomal 9AC-G$^W$ in plasma samples was about 350-fold greater than free 9AC-G$^W$. These results are consistent with stable retention of 9AC-G$^W$ in liposomes in vivo. The glycosidic switch may also increase safety since any drug leaking from the liposomes is in a highly hydrophilic form that is rapidly excreted.

We next asked if 9AC could be regenerated within tumors after intravenous administration of 9AC-G$^W$ liposomes. HPLC measurement of parental 9AC in tumor and blood samples demonstrated that the concentration of 9AC was over 30-fold higher in tumors than in blood at 24 h after i.v. injection of 9AC-G$^W$ liposomes (Fig. 9b). At 72 h post-injection, the amount of 9AC in tumors was about 12-fold higher than in blood, further confirming that the lysosomal-responsive trigger functions

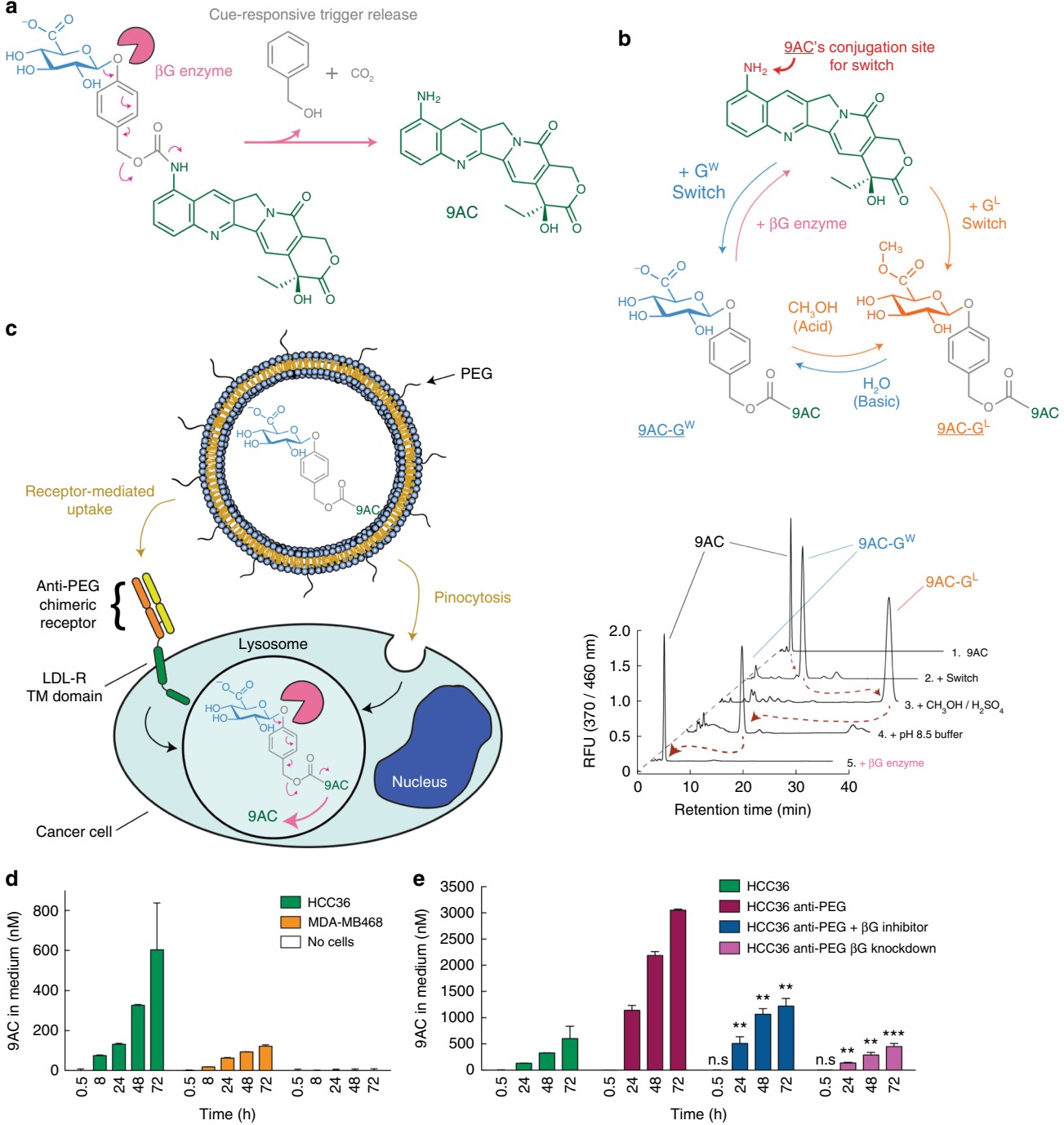

**Fig. 7** Environmental and cue-responsive behavior of the glycosidic switch. **a** Mechanism of cue-responsive triggering by lysosomal enzyme beta-glucuronidase (shown in pink). Enzymatic digestion triggers the degradation of the linker (shown in grey) and release of the parental drug 9AC (shown in green). **b** Illustration (upper panel) and HPLC analysis (lower panel) of the conversion processes between the different forms of the glycosidic switch using 9AC as an example. HPLC peaks showing conversion of 9AC to 9AC-G$^W$ (by chemical synthesis), switching to 9AC-G$^L$ in acidic methanol, reconversion to 9AC-G$^W$ in the pH 8.5 liposomal internal buffer and regeneration of 9AC by human beta-glucuronidase. RFU relative fluorescent units. **c** Illustration of cells engineered to express anti-PEG/LDL transmembrane domain (TM domain) chimeric receptors for enhanced endocytosis of PEGylated liposomes followed by triggered release of drug by beta-glucuronidase in lysosomes. **d** Accumulation of 9AC in the cell culture medium after addition of 9AC-G$^W$ liposomes to HCC-36 or MDA-MB468 cells or to empty wells without cells. **e** Accumulation of 9AC in the culture medium after addition of 9AC-G$^W$ liposomes to HCC36 cells or HCC36 cells expressing anti-PEG chimeric receptors. HCC36/anti-PEG cells were also treated with beta-glucuronidase inhibitor or beta-glucuronidase shRNA knockdown before addition of liposomes. Error bars: SD, $n = 3$. Statistical significance of differences in mean values: n.s not significant, $^{**}p < 0.001$ and $^{***}p < 0.0001$

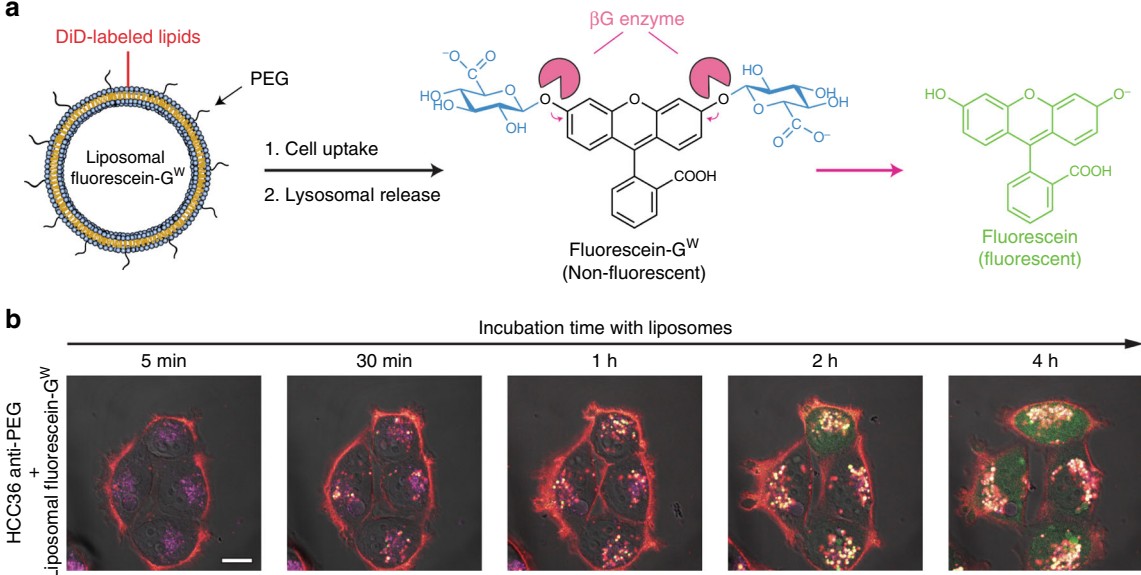

**Fig. 8** Confocal imaging of the intra-cellular regeneration of fluorescein from liposomal fluorescein-$G^W$. **a** Schematic representation of liposomal fluorescein-$G^W$ uptake by cancer cells and lysosomal beta-glucuronidase digestion of the glycosidic switch. **b** Regeneration of fluorescein from liposomal fluorescein-$G^W$ in HCC36 cells expressing anti-PEG chimeric receptors observed by confocal microscopy. Images show four representative cells, red indicates DilC18(5)-labeled liposomes, magenta indicates lysosomes, and green indicates fluorescein. Representative scale bar for all image = 10 μm

correctly after uptake of 9AC-$G^W$ liposomes in tumors in vivo. We speculate that cargos employing the lysosomal-responsive trigger may favorably synergize with liposome formulations that display enhanced endocytosis[40] or destabilization after endocytosis[42,43]. This approach might also benefit from lysosomal-specific targeting for enhanced regeneration of molecular cargos[44,45].

The in vivo antitumor activity of 9AC-$G^W$ liposomes was tested in a model of human breast cancer. MDA-MB-468 cancer cells were injected subcutaneously in immune-deficient NOD/SCID mice and allowed to grow until the average tumor size was between 75 and 100 mm³. The mice were then treated with four weekly i.v. injections of 14.5 μmol kg⁻¹ 9AC-$G^W$ liposomes, 14.5 μmol kg⁻¹ free 9AC-$G^W$, 5.5 μmol kg⁻¹ 9AC (the highest dose achievable due to poor drug solubility) or vehicle (TBS) and tumor growth, body weight, and survival were monitored (Supplementary Fig. 7a). MDA-MB468 tumors grow slowly in mice, which might more closely mimic the growth rate of tumors in humans, and may therefore be a more appropriate model of cancer than faster growing xenografts. We chose doxorubicin liposomes as the closest control for 9AC-$G^W$ liposomes, as both are stable long-circulating liposomal formulations of anticancer compounds. The chosen dosage was based on considerations of solubility and toxicity since we wished to test a condition representing a good compromise between therapeutic efficacy and toxicity. 9AC and free 9AC-$G^W$ had minimal effect on tumor growth as compared to TBS alone but 9AC-$G^W$ liposomes significantly suppressed tumor growth (Fig. 9c) and increased the survival rate of the mice (Supplementary Fig. 7b). All mice appeared to be tumor free except for one mouse, which displayed residual tumor mass (volume < 0.05 mm³) at 60 days after initiation of treatment with 9AC-$G^W$ liposomes. None of the treatments produced serious toxicity as monitored by total body weight loss (Fig. 9d). We also treated mice bearing the same tumors with Doxisome to get a sense of the relative efficacy of 9AC-$G^W$ liposomes in comparison to a clinically approved nanomedicine. Four weekly injections of liposomal doxorubicin (Doxisome at 1.8 μmol kg⁻¹) produced significantly less tumor

suppression as compared to treatment with 9AC-$G^W$ liposomes (Fig. 10a). Increasing the dose of Doxisome to 5.5 μmol kg⁻¹ created major toxicity and the death of all mice, which was not observed with 9AC-$G^W$ liposomes (Fig. 10b). The in vivo anticancer activity of 9AC-$G^W$ liposomes was surprisingly more effective than Doxisome; 9AC-$G^W$ liposomes cured tumors even though the difference in the in vitro IC$_{50}$ values of 9AC-$G^W$ liposomes and Doxisome was only about 20 fold (Supplementary Table 1). This result suggests that encapsulation of potent drugs in liposomes may be highly beneficial to help overcome inherent limitations in tumor drug accumulation afforded by the EPR effect. We believe that to counterbalance the low efficacy of EPR delivery, the potency of nanomedicines is a critical factor for successful tumor therapy. The recent demonstration of the benefits of nanoparticle drug delivery of a highly potent antiproliferative compound, monomethylauristatin E (MMAE), also supports this idea[46]. Since the potency of 9AC is about two orders of magnitude greater than doxorubicin, we suspect that the benefit obtained by 9AC-$G^W$ liposomes is significantly greater than with doxorubicin liposomes, as demonstrated by our in vivo results. In a similar way, early antibody-drug conjugates (ADCs) that employed unstable linkers to attach moderately potent anticancer agents such as doxorubicin or methotrexate produced disappointing clinical benefits[47,48], whereas later ADCs that employed stable linkers to attach highly potent drugs have displayed spectacular clinical anticancer activity[49,50]. We speculate that the excellent antitumor activity of 9AC-$G^W$ liposomes is therefore due to the high number of encapsulated drug molecules (~13,000 per liposome), their stable retention in the liposomes, which allows sufficient time for enhanced tumor accumulation, and the selective generation of a potent anticancer agent (9AC) inside cancer cells. In contrast to ADCs, liposomes do not require identification of tumor-associated antigens for selective tumor delivery. The glycosidic switch may therefore facilitate retention, delivery and safety of a large range of highly potent hydrophobic anticancer drugs that have hitherto been difficult to stably encapsulate in nanocarriers and provide more effective anticancer therapies.

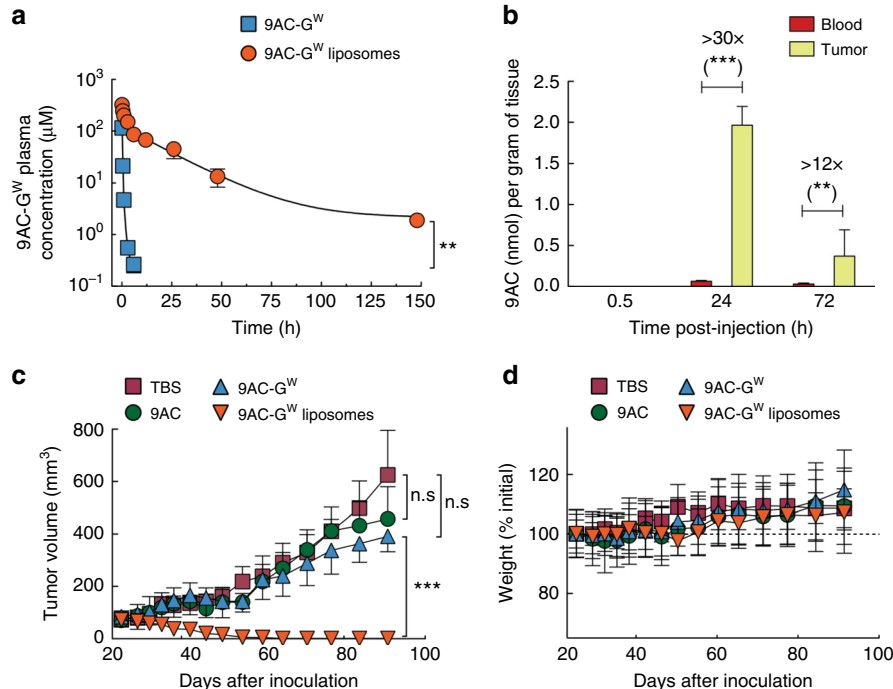

**Fig. 9** In vivo activity of 9AC-G$^W$ liposomes. **a** Concentrations of total 9AC-G$^W$ in plasma (free plus liposomal) at the indicated times after i.v. injection of 35 µmol kg$^{-1}$ 9AC-G$^W$ or 9AC-G$^W$ liposomes in mice. Error bars: SD, $n = 3$. **b** The concentration of 9AC in blood and MDA-MB-468 tumors after i.v. injection of 14.5 µmol kg$^{-1}$ 9AC-G$^W$ liposomes. Error bars: SD, $n = 3$. Statistical significance of differences in mean values are indicated: **$p < 0.001$ and ***$p < 0.0001$. The mean tumor volume (**c**), and mean body weight (**d**) of the treated mice bearing MDA-MB-468 tumors, injected i.v. four times at weekly intervals with TBS (control), 9AC (5.5 µmol kg$^{-1}$), 9AC-G$^W$ (14.5 µmol kg$^{-1}$) or 9AC-G$^W$ liposomes (14.5 µmol kg$^{-1}$) are shown. Error bars: SD, $n = 7$ mice per group. Error bars: SD. Stars indicate significance; n.s non-significant, **$p < 0.001$ and ***$p < 0.0001$

## Methods

**Reagents**. DSPC, 1,2-dioleoyl-sn-glycero-3-phosphocholine (DOPC), 1,2-distearoyl-sn-glycero-3-phosphoethanolamine-N-[methoxy(polyethylene glycol)-2000] (DSPE-PEG-2000), 1,2-dioleoyl-sn-glycero-3-phosphoethanolamine-N-[methoxy(polyethylene glycol)-2000] (DOPE-PEG), and cholesterol were purchased from Avanti Polar Lipids, Inc. (Alabaster, AL). BQC was a generous gift from Daiichi (Tokyo, Japan). 9AC[51], 9-aminocamptothecin-β-D-glucuronide (9AC-G$^W$), and 5,6-dihydro-4H-benzo[de]quinoline-camptothecin-β-D-glucuronide (BQC-G$^W$) were chemically synthesized. Briefly, the glycosidic switch was coupled to 9AC via a self-immolative carbamate linker[24] and to BQC via a self-immolative benzyl-ether spacer[25]. 4-methylumbelliferyl-β-D-glucuronide (4MU-G$^W$), beta-glucuronidase inhibitor (saccharolactone) and lysotracker-red DND99 were from Sigma Aldrich (St. Louis, MO). Fluorescein-di-glucuronide (fluorescein-G$^W$, F2915) was from ThermoFisher Scientific. PEGylated liposomal doxorubicin (Doxisome) was a generous gift from the Taiwan Liposome Company (TLC, Taipei, Taiwan).

**Cell lines and animals**. LS174T, Colo205, and SW620 human colon adenocarcinoma, HT29 human colorectal adenocarcinoma, HCT116 human colorectal carcinoma, NCI-H2170, and SK-MES-1 human lung squamous cell carcinoma and MDA-MB-468 breast adenocarcinoma cells were obtained from the American Type Culture Collection (ATCC, Manassas, VA, USA). CL1-5 human lung adenocarcinoma cells were kindly provided by Dr. Pan-Chyr Yang (Academia Sinica, Taipei, Taiwan)[52] and LS174T human colon adenocarcinoma high-mucin variant HM7 cells were kindly provided by Dr. Kyu Lim (Chungnam National University, Korea). HCC36 human hepatocellular carcinoma cells stably expressing a chimeric anti-PEG (HCC36 anti-PEG) receptor, were produced by replacing the extracellular part of the low density lipoprotein receptor (LDL-R) by an anti-PEG antibody to form an endocytic anti-PEG/LDL-R receptor[53]. Beta-glucuronidase protein production in selected cell lines was knocked down by a single-hairpin RNA (shRNA) expression plasmid (pLKO.1) targeting the 5′-CCGAATCAC-TATCGCCATCA sequence. The plasmid was obtained from the National RNAi Core Facility (Academia Sinica, Taipei) and the transfection was achieved by recombinant lentivirus[54]. All cells were cultured in RPMI 1640 supplemented with 10% heat-inactivated FBS, 2.98 mg mL$^{-1}$ HEPES, 1 mg mL$^{-1}$ sodium bicarbonate, 100 units mL$^{-1}$ penicillin, and 100 µg mL$^{-1}$ streptomycin in a 5% CO$_2$ humidified atmosphere in air at 37 °C. All cell lines were tested for mycoplasma and propagated for less than 6 months. Twelve to 16 week old female NOD/SCID immune deficient mice were obtained from the National Laboratory Animal Center, Taipei, Taiwan. The animals were maintained under specific pathogen-free conditions. All

animal experiments were performed in accordance with the institutional guidelines and approved by the Laboratory Animal Facility and Pathology Core Committee of the Institute of Biomedical Sciences, Academia Sinica. The mice were randomly assigned to treatment groups but the investigators were not blinded during the treatments.

**Synthesis and purification of lipid-soluble drug conjugates**. Twenty five milligram of 9AC-G$^W$, BQC-G$^W$ or 4MU-G$^W$ were dissolved in 1 mL methanol with addition of 3.5 µL sulfuric acid (H$_2$SO$_4$) to act as catalysis for the generation of lipid-soluble 9AC-G$^L$, BQC-G$^L$ and 4MU-G$^L$. The solutions were incubated at 62 °C and the reactions were monitored by HPLC. Purification was used to remove H$_2$SO$_4$ using a hand-packed preparative column of LiChroprep® RP-18 (40–63 µm) (Merck, Germany) equilibrated with 5% methanol in water. 9AC-G$^L$, BQC-G$^L$, and 4MU-G$^L$ were eluted with 100% methanol at 10 mL min$^{-1}$, dried under rotary evaporation and suspended in DMSO. Final concentrations were determined by analytical HPLC and adjusted to 10 mg mL$^{-1}$.

**Water solubility and octanol-to-water partitioning**. Water solubility was determined by dissolving compounds in water until they reached saturation, equilibration for 24 h with shaking at room temperature, and centrifugation at 20,000 × $g$ for 30 min at 20 °C. Samples from the supernatant were analyzed by HPLC to determine water solubility. The partitioning of compounds between octanol and water was determined by adding 2.5 nmol of each substance to an equal volume (200 µL) mixture of octanol and 0.25 M citric acid, pH 5 and shaking for 24 h to permit equilibration between the two phases. The tubes were then centrifuged at 20,000 × $g$ for 30 min at 20 °C and 20 µL samples from each phase were analyzed by HPLC. The partition coefficient is defined by the following equation, where "$C$" is the concentration of compound in each phase. A partition coefficient equal to 0 means that the compound equally distributes between each phase. Higher than 0 means that the compound preferentially distributes in octanol while less than 0 means the compound preferentially distributes in water.

$$\text{Partition coefficient} = \text{Log}\left(\frac{C[\text{octanol}]}{C[\text{water}]}\right) \qquad (1)$$

**Liposome preparation**. DSPC, DSPE-PEG-2000, and cholesterol were dissolved in chloroform at a 65:5:30 molar ratio, respectively. A dried lipid film was formed at 65 °C by rotary evaporation (Büchi, Rotavapor RII) and rehydrated with 250 mM calcium acetate, 50 mM HEPES pH 8.5 at 65 °C to a final lipid concentration of 10

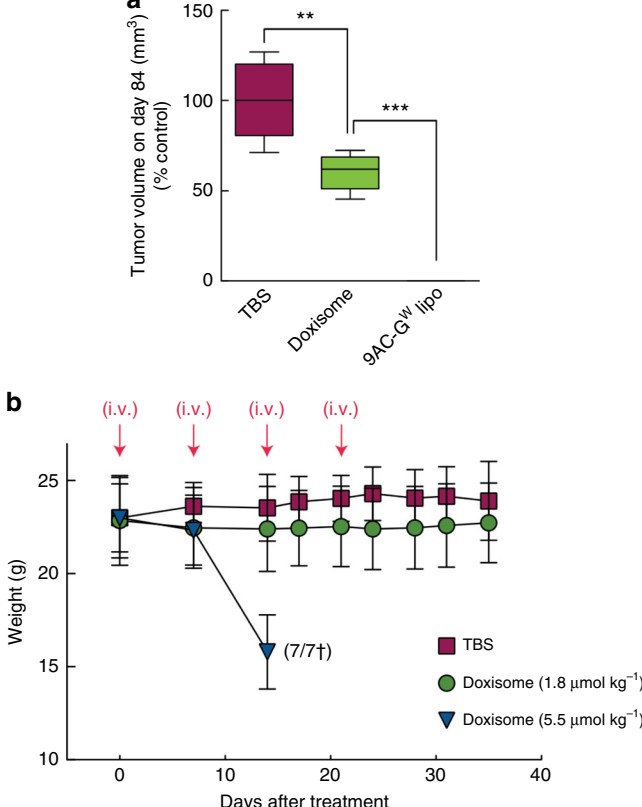

**Fig. 10** In vivo antitumor activity and toxicity of doxorubicin liposomes (Doxisome). **a** Mean tumor volumes in mice bearing subcutaneous MDA-MB-468 human breast tumors at day 84 after tumor inoculation. Results are shown as percentages relative to the control group (TBS) of mice treated weekly four times with Doxisome (1.8 μmol kg$^{-1}$) or 9AC-G$^W$ liposomes (14.5 μmol kg$^{-1}$). **b** Mean weights and survival of mice treated with Doxisome at 1.8 μmol kg$^{-1}$ or 5.5 μmol kg$^{-1}$. Error bar: SD, $n = 7$. Statistical significance of differences in mean values: **$p < 0.001$ and ***$p < 0.0001$

mg mL$^{-1}$. The liposomal suspension was submitted to ten freeze/thaw cycles using liquid nitrogen and a water bath at 70 °C, followed by 21 extrusions at 72 °C through 400, 200, and 100 nm polycarbonate membranes using a mini-extruder (Avanti Polar Lipids, Inc.). The liposomal external medium was changed by size exclusion column chromatography on Sephadex® G50 (Sigma Aldrich) to 250 mM sodium sulfate, 50 mM citric acid, pH 5. The pH of the hydration and exchange buffers were adjusted with 10 N NaOH. The resulting ready-to-load liposomes were stored at 4 °C for a maximum of 1 month before loading. The final lipid concentration was determined by Bartlett's phosphorus assay[55] and the average diameter of the particles was measured by dynamic light scattering on a Zetasizer Nano system (Malvern Instruments).

**Drug loading**. Liposomes (stock: 6.5 μmol mL$^{-1}$ or 5 mg mL$^{-1}$), and drugs (stock: 15 μmol mL$^{-1}$ in DMSO) were pre-warmed separately at 75 °C for 10 min before mixing at 1:4.6 drug-to-lipid molar ratios for 9AC and BQC based compounds, and at 1:2.3 for 4MU based compounds. All drug stock solutions were prepared in DMSO to prevent undesired chemical hydrolysis during storage and to maintain a constant DMSO concentration of 9% (v:v) in the loading buffer for both water-insoluble parental drugs and water-soluble glycosidic switch drugs. The drug loading was performed at 75 °C for 1 h with occasional gentle hand shaking. The liposomes were then transferred to ice for 10 min and non-encapsulated drug was removed on a Sephadex® G50 column equilibrated with filtered 50 mM Tris-Cl, 150 mM NaCl adjusted to pH 7.4 (Tris-buffered saline, TBS). Loading efficiency, saponification yield, and final drug-to-lipid molar ratios was determined by HPLC after dilution of the liposomes in 25 mM citric acid pH 2.9 containing 1% Triton X-100.

**Membrane thickness measurement**. Micrographs obtained from cryogenic electron microscopy were enlarged to their actual size in ImageJ (NIH). The amount of pixels corresponding to the 100 nm scale bar was determined using the "Set Scale" tool, prior to randomly measuring the membrane thickness at different positions of either empty liposomes or liposomes loaded with 9AC, 9AC-G$^W$, BQC or BQC-G$^W$. The randomization was obtained by overlapping a grid over the micrographs and thickness was measured each time the grid contacted with a liposomal membrane, until 40 values were collected.

**HPLC analysis**. HPLC analysis was performed using a reverse phase C18 column (Waters, μbondapak™ C18, 3.9 × 300 mm, 10 μm). 9AC, 9AC-G$^W$, 9AC-G$^L$, 4MU, 4MU-G$^W$, and 4MU-G$^L$ were separated at 2 mL min$^{-1}$ in a mobile phase composed of 24% acetonitrile in a pH 2.9 buffer composed of 25 mM citric acid adjusted by 10 N NaOH. For all analysis, camptothecin drugs were incubated at pH 2.9 before injection to allow the reformation of the lactone ring. For the separation of BQC, BQC-G$^L$, and BQC-G$^W$, 30% acetonitrile was used. All mobile phases were degassed by sonication under vacuum and filtered through a 0.2 μm bottle top filter membrane (Nalgene® Rapid-Flow). Detection was performed with a fluorescence detector (Jasco FP-2020) at excitation/emission wavelengths of 370/460 nm for 9AC, 9AC-G$^W$, 9AC-G$^L$, BQC-G$^W$, BQC-G$^L$, and 370/575 nm for BQC. 4MU, 4MU-G$^W$, and 4MU-G$^L$ were detected with a U.V. detector (Jasco U.V. 975) at 310 nm. Data was collected and analyzed on Gold software (Beckman).

**Cryo-transmission electron microscopy**. Holey carbon film-covered 400-mesh copper grids (HC300-Cu, PELCO) were glow-discharged in an argon and oxygen atmosphere for 10 s on the carbon side. A volume of 4 μL of liposomes (2 mg mL$^{-1}$) was pipetted onto the grids, blotted in 100% humidity at 20 °C for 3 s and plunge-frozen into liquid ethane cooled by liquid nitrogen using a Vitrobot (FEI, Hillsboro, OR). Grids were then stored under liquid nitrogen and transferred to the electron microscope using a cryostage. Images of liposomes within the holes in the carbon film were obtained on a Tecnai F20 electron microscope (FEI) at 200 keV with a 70 nm objective aperture. The low dose condition for each exposure was ~20 e$^{-}$ (Å$^2$)$^{-1}$. Images were taken at 5 k or 50 k magnification and 2–3 nm defocus and recorded on a 4k × 4k CCD camera (Gatan, USA).

**In vitro release kinetics**. Liposomes were diluted at 1.3 μmol mL$^{-1}$ (1 mg mL$^{-1}$) of total lipid in 2 mL PBS supplemented with 10% FBS and transferred to Pur-A-Lyser Maxi 12000 dialysis tubes (Sigma-Aldrich). The dialysis was run at 37 °C under stirring in 2 L of PBS containing kanamycin (50 μg mL$^{-1}$) and carbenicillin (100 μg mL$^{-1}$) to prevent bacterial growth during the experiment. Samples (5 μL) were taken at different times from the liposomal fraction and directly diluted in 195 μL of HPLC mobile phase with 1% Triton X-100 to analyze the total remaining amount of encapsulated drug by HPLC.

**Reversibility of the glycosidic switch**. Water-soluble 9AC-G$^W$ was obtained by incubating 50 μg 9AC-G$^L$ in 500 μL of liposomal internal buffer at pH 8.5. After 1 h at 65 °C the reaction was analyzed by HPLC to assess the reformation of 9AC-G$^W$. A 20 μL sample of 9AC-G$^W$ was then transferred to 1 mL of reaction buffer (100 mM sodium acetate, 100 ng mL$^{-1}$ bovine serum albumin, pH 4.5) containing 1 μg of purified recombinant human beta-glucuronidase[56] for 1 h at 37 °C to cleave the cue-responsive trigger and generate 9AC. Drug regeneration was also examined in a biological system where 10$^5$ cells (MDA-MB468, HCC36, and HCC36 anti-PEG) were seeded per well in a 96 well-plate in triplicate and incubated overnight. 9AC-G$^W$ liposomes were added to the cells at 20 μM (9AC-G$^W$ concentration) and 100 μL of medium was harvested after 30 min, 8 h, 24 h, 48 h and 72 h. In some experiments, 100 μM saccharolactone (a beta-glucuronidase inhibitor) was added with the liposomes. An equal volume (100 μL) of 2% Triton X-100 in 20% acetonitrile, 25 mM citric acid pH 2.9 was added to each sample and 50 μL were injected in the HPLC for analysis. In a confocal microscopy experiment, direct visualization of the reversibility of the glycosidic switch was performed by seeding 5 × 10$^5$ HCC36 anti-PEG cells 2 days prior to imaging in 6 well-plates containing 30 mm coverslip. Cells were stained with Lysotracker-red DND99 (75 nM) to stain lysosomes, diluted in RPMI medium supplemented with 10% BCS in a 5% CO$_2$ humidified atmosphere, and incubated for 30 min at 37 °C. The cells were washed twice with fresh medium and DiIC18(5)-labeled liposomes composed of 64% DOPC, 5% DOPE-PEG, 30% cholesterol and 1% DiIC18(5), loaded with fluorescein-di-glucuronide (fluorescein-G$^W$), were added to the cells at 160 μg mL$^{-1}$ of lipid. The loading of fluorescein-G$^W$ was achieved by mixing liposomes with fluorescein-G$^L$ prepared in a fashion similar to 9AC-G$^L$ and BQC-G$^L$. Images were collected on a Zeiss LSM780 laser scanning microscope (Carl Zeiss AG, Germany) and analyzed with Fiji software and ImageJ (NIH).

**Pharmacokinetics**. 9AC-G$^W$ was dissolved at 10.9 μmol mL$^{-1}$ (7.5 mg mL$^{-1}$) in 25 mM phosphate buffer pH 6.5 just before use. 9AC-G$^W$ liposomes were concentrated by successive centrifugations at 5000, 10,000, and 20,000 × $g$ for 30 min at 4 °C. After each centrifugation, the pellet was collected and resuspended in a smaller volume of buffer. The concentration of liposomal 9AC-G$^W$ achieved through this method before injection was 5.1 μmol mL$^{-1}$ (3.5 mg mL$^{-1}$) based on 9AC-G$^W$. Each preparation (free 9AC-G$^W$ and liposomal 9AC-G$^W$) was injected intravenously in the tail vein of 12–16 week old NOD/SCID female mice at 35 μmol kg$^{-1}$ (25 mg kg$^{-1}$) using three mice per group. Blood samples were collected at 5 min, 30 min, 60 min, 2 h, and 3 h for free 9AC-G$^W$, in addition to 6, 12, 26, 48,

148 h for 9AC-G$^W$ liposomes with Na-heparinized hematocrit tubes from the tail vein. The plasma was separated from the red blood cells by centrifugation at 2500 × $g$ for 3 min. Plasma samples containing liposomes were destabilized with 1% Triton X-100 to lyse the liposomes and diluted 1:1 (vol:vol) with acetonitrile: methanol:0.2 M trichloroacetic acid (2:2:1, vol:vol) to precipitate proteins[57] and centrifuged at 20,000 × $g$ for 5 min at 4 °C. The supernatant was directly injected into a C18 HPLC column with fluorescent detection at 370/460 nm (Ex/Em). The final results are expressed as 9AC-G$^W$ plasma concentration in μmol L$^{-1}$ (μM). Biological half-lives were determined by an exponential two phase decay curve fit on GraphPad prism (GraphPad software, Inc.).

**Regeneration of parental drug within tumors**. Twelve to 16 week old female NOD/SCID immunodeficient mice were inoculated with 10$^7$ MDA-MB468 human breast cancer cells. When tumors reached ~100 mm$^3$, the mice were intravenously injected with 14.5 μmol kg$^{-1}$ (10 mg kg$^{-1}$) of liposomal 9AC-G$^W$ (based on the drug concentration). Blood was collected in Na-heparinized hematocrit tubes from the tail vein and mice were then killed to harvest the tumor tissue. Blood samples (100 μL) were centrifuged at 2500 × $g$ for 3 min and the plasma was collected for analysis. Tumor tissue was homogenized in 3 mL RIPA buffer (25 mM Tris-HCl pH 7.5, 150 mM NaCl, 0.1% SDS, 0.5% Na-deoxycholate, 1% Triton X-100) on a IKA Ultra-Turrax® TP 18/10 homogenizer. The amount of 9AC was determined in blood and tumor samples by HPLC and normalized to amounts per gram of tissue, according to the relationship 1 mL of whole blood = 1.06 g of weight.

**In vitro proliferation assay**. The cells proliferation was measured as described in Supplementary Methods.

**In vivo anticancer activity**. MDA-MB468 human breast cancer cells (10$^7$ cells in 100 μL PBS) were injected subcutaneously in twelve to sixteen-week old female NOD/SCID immunodeficient mice. When tumors reached sizes ranging from 75 to 100 mm$^3$, groups of seven mice were treated at weekly intervals for four times with intravenous injections of 9AC at 5.5 μmol kg$^{-1}$ (2 mg kg$^{-1}$), 9AC-G$^W$ at 14.5 μmol kg$^{-1}$ (10 mg kg$^{-1}$), liposomal 9AC-G$^W$ at 14.5 μmol kg$^{-1}$ (10 mg kg$^{-1}$), or doxorubicin liposomes (Doxisome, 1.8 μmol kg$^{-1}$ equivalent to 1 mg kg$^{-1}$ and 5.5 μmol kg$^{-1}$ equivalent to 3 mg kg$^{-1}$). Tumor volumes, body weights, and survival were monitored for more than 2 months after treatments started. Mice were sacrificed when their lower limbs exhibited signs of impairment, generally before the external visible part of tumors exceeded 1000 mm$^3$. The tumor volume was calculated according to the following formula:

$$\text{Tumor Volume} = \frac{\text{Length} \times \text{Depth} \times \text{Height}}{2} \qquad (2)$$

**Statistical analysis**. Results are presented as the mean, plus or minus (±) standard deviation (S.D.). All experiments were repeated at least two times with representative data shown. Animal sample size was chosen based on similar well-characterized literature. Statistical analyses were examined using the two-tailed unpaired Student's $T$-test. Differences in tumor sizes between groups were examined for statistical significance using one-way ANOVA followed by Dunnett's multiple comparisons; a $p$ value of less than 0.05 was considered significant. No statistical method was used to predetermine sample sizes.

**Data availability**. The data that support the findings of this study are available from the corresponding authors upon reasonable request.

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

## Acknowledgements

This work was supported by grants from Academia Sinica (AS-107-TP-B11 and Research Program on Nanoscience and Nanotechnology) and the Ministry of Science and Technology, Taiwan (NSC101-3111-Y-001-008). We appreciate Dr. Yuan-Chih Chang and H. J. Huang for assistance in the use of the Tecnai F20 in the Cryo-EM Core Facility, Scientific Instrument Center at Academia Sinica. The authors thank Dr. Shu-Chuan Jao of the Biophysics Core Facility, Scientific Instrument Center at Academia Sinica, Taipei, Taiwan for assistance in performing dynamic light scattering on a Zetasizer Nano system (Malvern Instruments). We thank Ms. Show-Rong Ma of the confocal Microscopy Core Facility, Scientific Instrument Center at the Institute of Biomedical Sciences, Academia Sinica, Taipei, Taiwan, for helping with the Zeiss LSM780 laser-scanning microscope (Carl Zeiss AG, Germany). We also thank the Taiwan Liposome Company, Taipei, Taiwan for generously providing Doxisome.

## Author contributions

P.-A.B and S.R.R. conceived the project and designed the experiments. P.-A.B. performed the experiments. P.-A.B. and Y.-C.S. performed the in vivo antitumor activity experiment. P.-A.B. and K.W. performed experiments examining in vitro regeneration of the parental drug and confocal imaging. W.-C.L. and Y.-L.L. synthesized 9AC-G$^W$ and BQC-G$^W$. P.-A.B. wrote the manuscript with help, suggestions and editing from S.R.R. All authors approved the final version of the manuscript.

## Additional information

**Competing interests:** The authors declare no competing interests.

