## [Peer Review File · Nature Communications]

Editorial Note: This manuscript has been previously reviewed at another journal that is not operating a transparent peer review scheme. This document only contains reviewer comments and rebuttal letters for versions considered at Nature Communications. Mentions of prior referee reports have been redacted.

Reviewers' Comments:

Reviewer #6:

Remarks to the Author:

The manuscript has intriguing approach of hydrophilic-lipophilic switch enabled loading of anticancer drugs in the liposome aqueous core. Extensive evaluation of the formulation has been carried out to establish the proof of concept. Authors' have also addressed the previous reviewer's comments in the revised manuscript. However, there are following comments that need to be resolved before publication.

1. Camptothecins have been noted to be very unstable in aqueous media due to their closed lactone ring which opens at alkaline pH ($\text{pH} > 5.5$) forming inactive open lactone form. The rates of hydrolysis of CPT in a PBS ($\text{pH} 7.4$) have been studied and it has been shown that only 10% activity remained after 2 hr in PBS. Moreover, liposome development has been targeted to stabilize the compounds in the bilayer where its lactone is protected from the hydrolytic aqueous phase. Keeping these in mind and use of $\text{pH} 8.5$ inside liposomes, I think authors should describe what actually they are measuring in the liposomes i.e. is it the of ACG closed lactone or ACG open lactone? The explanation makes it confusing, as it could be a mixture of components inside such as 9AC-GW open lactone, 9AC-GW closed lactone, calcium salts of lactone, calcium salt of glucuronate etc.). This in turns requires explanation for different precipitation phenomenon inside the liposomes.

2. Authors showed that the switch molecules are hydrolyzed to water soluble counterparts (Fig 2 Conversion of 9AC-GL to 9AC-GW), however, they never emphasized on the fact that it is not actually the conversion of 9AC-GL to 9AC-GW and rather it is conversion to open lactone 9AC-GW. Authors need to bring this to notice in the manuscript and also explain the disposition occurring in the cancer cells under acidic conditions which is essential for activity of 9AC (about lactone ring).

3. This in turn also warrants the explanation for further disposition and activity of other molecules (glucuronic acid, calcium and benzyl alcohol) generated at equimolar or higher concentrations as drug after the glucuronidase cleavage. Don't these molecules impart their effects on the cytotoxicity or other effects?

4. Secondly, when authors have used methanol for esterification, this makes it important that authors explain the release of methanol upon hydrolysis. As methanol could be toxic to the cells, they need to rationally justify based on the quantity of the methanol that the cells will be exposed upon dosing and give relevant deduction.

5. Authors' generalization to apply the switch methodology to a wide range of compounds is highly speculative. Even though 9AC, BQC and 4MU had similar log P values, their switch versions i.e. W and L versions gave different partitioning to the molecules with logP of 0 for 4MU-GL which very low as compared to logP of 9AC-GL and BQC-GL. This important factor needs to be considered to make generalization. Moreover, the application is only possible due to the lactone ring of the aminocamptothecin which is an important factor to consider for the switch based modification of molecule otherwise it is just another molecule as doxorubicin. This in turn makes the choice of 4MU improper.

6. Authors responded to the reviewer 2's comment on the dose selection of free and liposomal formulation. Though authors tried to justify the dose difference due to limitations of solubility, it would have been better to estimate the effectiveness at the same dose level as in clinical settings, iv anticancer drugs are always given by infusion which to some extent bypasses the solubility limitation. Anyhow, the dosing is also possible in the fractions, if authors had considered that.

7. Authors have given the in vivo activity of 9AC-GL liposomes against the free drug and parental drug (Fig 6). The results show that there is an unprecedented therapeutic activity of the liposomes which doesn't even allow the tumor to grow. Further, in Supplementary fig 10 a, the difference in tumor size between the Doxisome and 9AC-GL liposomes is unbelievable (~ 65 vs 0; i.e. > 60 times). However, this doesn't coincide with the IC50 assay in which IC50 values are (2.4 vs 0.15, i.e. > 24 times). Usually, the in vivo studies turn out to give results which most probably are lower as compared to in vitro and at the highest luck, same as in vitro results. The results need appropriate justification.

8. Release studies of the liposomes were performed in PBS. These studies need to be performed in three mediums (pH 7.4, pH 6.4 and pH 5.5) to represent the uptake of liposomes in the normal tissue, cancer interstitium and inside cancer cells. Reported release study is sufficient to justify the retention of the drugs in liposomes during circulation.

9. As authors also agree with the fact that EPR doesn't ensure total delivery of the nanocarrier to tumors, yet it is the only mechanism which is primarily important for preferential accumulation of nanoparticles in tumors as opposed to other normal tissues due to the leaky vasculature and inefficient lymphatic drainage in tumors. However, distribution to other normal organs is what cannot be avoided completely. Hence, given the very low IC50s of the compounds, authors should consider a few in vitro studies to evaluate the cytotoxicity to normal cells i.e. liver cells, normal tissue cells (representative of the site of the cancer).

Response to reviewers

We wish to thank the reviewers for their careful examination of our revised manuscript. Their comments and suggestions have helped to improve the clarity and overall quality of our paper.

To help the reviewing process, we labeled in:

- BLUE, our responses to the reviewers.

- RED, the modifications added to our manuscript.

- GREEN AND UNDERLINED, are the comments copied from previous reviewers.

Reviewer #5

We thank reviewer #5 for helping us to improve our manuscript. Please find our responses to the expressed concerns below:

1. The addition of only 2 references mentioned by reviewer #1 did not sufficiently address the problem regarding the novelty and needs to be expanded.

To clarify the novelty brought by the present work, we added the following text within the introduction:

Additional information can be found at:

Page 3, line 59: For example, docetaxel was modified by a N-methylpiperazinyl butanoic acid group to produce a protonated derivative to allow the use of a pH gradient for improved loading²⁰. In another approach, docetaxel was modified by attachment of a glucose group to increase water solubility and improve loading capacity in liposomes by avoiding membrane accumulation of docetaxel²¹. Although glycosylated docetaxel retained 90.9 % of tubulin stabilization, such modification can strongly affect drug potency in other cases.

Page 4, line 73: The method may be a more general approach as we demonstrated the utility of the switch concept by stably retaining chemically different hydrophobic drugs in liposomes.

2. Reviewer #2 comments on the figures clarity and number were not sufficiently addressed

Thank you for your comment. We tried to make the figures as clear and simple as possible, while retaining important data on the main figures without needing to frequently check the supplementary data. We modified the figures as follows:

To help the reviewing, we wish to summarize here the changes that were made to the figure since our first version:

Figure 1: We simplified the overall concept. The figure was split between a) and b) panels. Panel a) shows the general concept and panel b) shows in more details the case of 9AC-G^W discussed within the paper. We removed unnecessary details.

Figure 2: We reduced the overall size of the figure to economize on space.

Figure 3: Panels a), b), e), f) and g) were moved to supplementary data. The diagram of panel h) (renamed as panel c)) was modified to improve clarity and save space. We also added experimental results of drug loading after modification with the glycosidic switch under a hydrophilic form in panel c.

Figure 4: We removed panel a) (previously showing cryo-EM images of empty liposomes, this information can already be found in the supplementary figure 1) and moved panel d) (showing a diagram of the drug release experimental setup) to supplementary figure 4. The

retention data graphs were combined (previously separated between parental drugs and drugs modified with the switch) resulting in better clarity for the reader and space saving.

Figure 5: We rearranged the organization of the panels for clarity and moved the fluorescein diagrams and data in panels c) and f) to supplementary figure 7, panels a) and b).

Figure 6: Panels c) (showing a diagram of the drugs treatment schedule) and e) (showing the survival of the mice after treatment) were both moved to supplementary figure 9, panels a) and b).

Some additional changes that we made to simplify our figures are listed below:

Figure 2, panel a.: Since it is not necessary to repeatedly show the parental structures of 9AC, BQC and 4MU after conjugation of the glycosidic switch, we replaced the structures of the parental drugs in 9AC-G^W, BQC-G^W and 4MU-G^W by simply labelling “9AC”, “BQC” and “4MU”. This helped to reduce the complexity of the figure.

Previous figure 2:

New figure 2:

Figure 2. Effect of switch conjugation on hydrophobic cargos. (a) The switch was conjugated to 9-aminocamptothecin (9AC) and 5,6-dihydro-4H-benzo[de]quinoline-camptothecin (BQC) via a cue-responsive trigger^{17, 24} to generate 9AC-G^W and BQC-G^W, while it was directly attached to 4-methylumbellirone to generate 4MU-G^W (The red arrows indicate the position where the glycosidic switch was conjugated). The carboxylic acid function of the switch is converted in acidic methanol to a lipid-soluble methylester, which can be reconverted to the water-soluble form in basic aqueous solutions. (b) Comparison of the water solubility of the parental drugs 9AC and BQC with their water-soluble switch conjugated forms (9AC-G^W and BQC-G^W). Error bars, SD, $n = 3$. (c) Kinetics of the conversion of 9AC-G^W to lipid-soluble 9AC-G^L, (d) BQC-G^W to BQC-G^L and (e) 4MU-G^W to 4MU-G^L in acidic methanol at 62°C. The HPLC chromatograms show samples at 0, 15, 30 and 60 minutes. (RFU = relative fluorescent units). (f) Partition coefficients (Log scale) between 50 mM citric acid buffer pH 5 and 1-octanol of the parental compounds compared to their respective -G^W and -G^L forms. Error bars: SD, $n = 5$. (g) Stability of 9AC-G^L in

liposomal external buffer (50 mM citric acid, 250 mM sodium sulfate, pH 5) at 75°C. Error bars: SD, $n = 3$. **(h)** Conversion of 9AC-G^L to 9AC-G^W in liposomal internal buffer (50 mM HEPES, 250 mM calcium acetate, pH 8.5) at 75°C. Error bars: SD, $n = 3$. Statistical significance of differences in mean values: $p < 0.0001$ (***)

Next we modified Figure 3 by removing the schematics in panel a and panel b.

Previous figure 3:

New figure 3:

After modification, we believe that all of the current data displayed on the figures are necessary for the understanding of the fundamentals.

- Reviewers # 1, #2 and #4 concerns were well addressed in the response letter but not in the manuscript, this needs addressing, especially for most of the fundamental and important questions and concerns contributed by reviewer #4.

Thank you, we have listed below the changes we made to more completely respond to the reviewers comments.

Reviewer #1:

“2. If the drug solubility of 9-AC could be significantly improved to 30-40 mg/mL, why does the drug stock solution need to be prepared in DMSO at 10 mg/ml?”

Drugs were dissolved in DMSO as stock solutions to prevent unwanted hydrolysis in water during storage. In addition, although the glycosidic switch drugs are soluble in water, the parental drugs are insoluble in water. We therefore dissolved all drugs in DMSO for a fair comparison and to remove the nature of the solvent as an experimental variable.

We modified the text to make this clear:

Page 8, line 147: All drug stock solutions were prepared in DMSO to prevent undesired chemical hydrolysis during storage and to maintain a constant DMSO concentration of 9% (v:v) in the loading buffer for both water-insoluble parental drugs and water soluble glycosidic switch drugs.

3. DMSO has been shown to increase the membrane permeability, thereby enhancing the drug loading efficiency. Thus, the increased loading efficiency might be attributed to the presence of DMSO over the lipophilicity of the drug. This needs to be discussed.”

For this comment, we only discussed in the manuscript the fact that we used the same concentration of DMSO for all conditions. We now added more details about the suspected improved loading efficiency that could be attributed to the presence of DMSO.

Additional information can be found at:

- Page 16, line 326: As DMSO is known to enhance the permeability of lipid bilayers, we used the same DMSO concentration (9%, v:v) for all the loading conditions to ensure that any improvement in drug loading efficiency is solely attributed to the influence of the glycosidic switch.

Reviewer #2:

“2. There is no direct evidence presented that the ester groups on the glucuronidated drugs are hydrolyzed inside the liposome. This must be shown. The release data in Figure 4D-F is suggestive, but not proof. Detailed new data to show this is needed.”

Data shown in Figure 3 and Supplementary Figure 2 provide direct evidence that the ester groups are hydrolyzed inside the liposomes. To make this clearer, we added the following sentence to the manuscript.

Additional information can be found at:

- Page 15, line 309: As the liposomal fraction was separated from the external medium and lysed with detergent to reveal its contents, this provides direct evidence that the ester group of the lipophilic switch was hydrolyzed inside the liposomes.

“4. The data in Figure 6d is intriguing, but rather flawed. Clearly, the liposomal preparation wins out. However, it is compared with free drug, which is administered at 1/5th the dose due to solubility considerations. Given the vast improvement in exposure that the authors allude to, they should have included 2 mg/kg of the liposomal preparation for direct comparison.”

Actually, on a molar base of active drug, the difference between liposomal and soluble drug concentrations was 2.6 fold. The dose of liposomal drug was selected because we did not want to minimize the possible efficacy of 9AC-G liposomes due to the limitations of administering free 9AC. The ability to administer more 9AC-G liposomes is an important advantage of this formulation. We included the reasons for this choice:

Additional information can be found at:

- Page 22, line 468: The chosen dosage was based on considerations of solubility and toxicity since we wished to test a condition representing a good compromise between therapeutic efficacy and toxicity.

“5. It is interesting that liposomal loading with the two camptothecins far exceeded that of 4MU. Why? What evidence do the authors have that the lactones in the camptothecin isn’t playing a role, given that the vast majority would be in the hydrolyzed at pH 8.5? Furthermore, there is no discussion of even why 4MU is included in the paper at all and what was learned from the small amount of attention this particular compound seemed to have garnered.”

We included 4MU in our manuscript to demonstrate the beneficence of the glycosidic switch for loading of compounds other than camptothecins. We believe that our strategy does not have to be restricted to a specific class of compounds like camptothecins, although the presence of the lactone ring might be beneficial. We believe that showing 4MU as an additional example provides an honest comparator as a model for non-camptothecin compounds. In addition 4MU might possess some interesting anticancer properties as mentioned in our manuscript.

We added some additional information at:

Page 13, line 267: 4MU was chosen as a representative of “non-camptothecin” compounds to demonstrate the utility of the glycosidic switch in a more general approach.

More discussion was added to explain the lower loading efficacy of 4MU-G as compared to 9AC-G and BQC-G.

Page 16, line 328: 4MU-G^L was loaded less efficiently than 9AC-G^L and BQC-G^L. We suspect that the partitioning of the modified drugs between the aqueous and the organic phases is important for efficient loading. For example, 4MU-G^L is only very slightly lipophilic (Log P ~ -0.002) and it appears that a clear partitioning towards the organic phase is preferred, as observed with BQC-G^L (Log P ~ 1.3) and 9AC-G^L (Log P ~ 1.1) for efficient loading. We suspect that esterification of the glycosidic switch with longer carbon chains (i.e. ethanol, propanol, etc.) might help to increase the log P values of such compounds and therefore the loading efficiency. In the end, the loading efficiency of 4MU-G^L was still superior to the parental compound 4MU (**Fig. 3c**).

Considering the effect of the lactone ring we also added the following:

Page 17, line 363: A beneficial effect of the camptothecin lactone ring on improved retention can also be considered, as the lactone ring is present in the open charged carboxyl form at the high pH inside liposomes. However, the poor retention of 9AC and BQC in the same liposomes (with high internal pH) demonstrates that opening of the lactone ring by itself is insufficient to achieve good retention of the drugs inside liposomes.

“8. The Doxisome comparison in Supp. Figure 3 seems a little odd, since the most obvious comparator would have been free drug. It’s a small point, but I’d like to know why the comparison was made the way it was.”

Additional information can be found at:

Page 19, line 416: We chose clinically-approved PEGylated liposomal doxorubicin (Doxisome) for comparison to examine the activity of 9AC-G^W liposomes relative to an established stable liposomal drug formulation that is used clinically for the treatment of patients suffering from ovarian cancer, AIDS-related Kaposi's sarcoma and multiple myeloma.

Reviewer #4:

“The underlying delivery principles are solely based on ‘enhanced permeation and retention (EPR)’ effect and free penetration of the extravasated liposomes into the tumor tissues to meet individual cancer cells for active or passive cellular uptake and conversion to the active drugs. Despite the countless reports of EPR effect in rodent cancer models, a few solid evidences support EPR effect in human patients. Clinical evidence of therapeutic EPR effect is even scarce. No evidence supports 130 nm liposome freely penetrates into the tumor tissues and no data support that the majority of individual cancer cells receives the liposomes. This may, in part, explain the reasons why multiple clinical trials of cancer nanomedicine become unsuccessful. Even a successful example of Doxil® (a similar example to this study—long circulating, stable, EPR effect, defined small size) in the market does not demonstrate better therapeutic efficacy when compared to free doxorubicin (J. Control. Release (2016) 232, 255-264).”

We agree that the EPR effect limits the amount of nanomedicine that can accumulate in tumors. For this reason, we suggest that the potency of the loaded compounds plays a major role in the success of anticancer drug delivery in therapy. Doxorubicin is not very potent, which may be a major reason that Doxil is only effective for tumors with exceptional EPR. We therefore added the following to the discussion:

Page 22, line 485: [...] encapsulation of highly potent drugs in liposomes may be highly beneficial to help overcome inherent limitations in tumor drug accumulation afforded by the EPR effect. We believe that in order to counterbalance the low efficacy of EPR delivery, the potency of nanomedicines is a critical factor for successful tumor therapy. The recent demonstration of the benefits of nanoparticle drug delivery of a highly potent antiproliferative compound, monomethylauristatin E (MMAE), also supports this idea¹. Since the potency of 9AC is about two orders of magnitude greater than doxorubicin, we suspect that the benefit obtained by 9AC-G^W liposomes is significantly greater than with doxorubicin liposomes, as demonstrated by our *in-vivo* results.

“There are no clinical evidences available supporting that long-circulating stable nano-sized delivery systems outperform to short circulating unstable nano-formulations. Examples contrast Doxil vs. Myocet and NK105 vs. Genexol PM.”

A recent study published in Nature Communications showed that stable nano-formulations bring beneficial effects compared to unstable nano-formulations (“*Augmenting drug-carrier compatibility improves tumor nanotherapy efficacy*”)². In this work, nanoparticles with poorly retained drug had low drug to tumor accumulation (~ 0.1 % ID) and were not effective to treat cancer. On the other hand, nanoparticles with strongly retained drug could deliver drug more efficiently to tumors (> 1 % ID) and had significantly better antitumor activity and survival. We added the new reference to the manuscript:

Page 3, line 55: This is important because stably retained drugs allow greater tumor accumulation than unstable drugs ($> 1\%$ ID vs. $\pm 0.1\%$ ID) and consequently display stronger anticancer activity and overall survival *in-vivo*².

“Technically, water-soluble drugs can be loaded in liposomes without active conversion from GL to G_w form during the hydration process.”

“The active loading may improve loading efficiency but compensated with chemistry involved. The reason for precipitation of G_w forms in the liposome is not clearly explained. Precipitation by the presence of divalent ions (Ca⁺⁺) is not supported. It is, then, not clear the slow release kinetics is due to precipitation or water-solubility.”

It is very difficult to directly load water soluble versions of the drugs as shown by the low loading of the water soluble versions in Fig. 3c. We speculate that calcium ions help retain the drugs by precipitation, but the actual mechanism requires further study. However, regardless of calcium-mediated effects or increased water solubility, clearly the glycosidic switch greatly increases the retention of 9AC and BCG inside the liposomes as shown in Figure 4. We believe that drug stability is more important than the effect on loading. We added the following text as explanations:

Page 17, line 345: The calcium ions in the liposomes likely form a complex with the glycosidic switch under a water soluble form (-G^W), by reaction between positively charged calcium and negatively charged carboxylate of the switch, leading to precipitation inside the liposomes, which may help retain molecular cargos during delivery³.

And more details were also provided about the potential effect of calcium:

Page 17, line 363: A beneficial effect of the camptothecin lactone ring on improved retention can also be considered, as the lactone ring is present in the open charged carboxyl form at the high pH inside liposomes. However, the poor retention of 9AC and BQC in the same liposomes (with high internal pH) demonstrates that opening of the lactone ring by itself is insufficient to achieve good retention of the drugs inside liposomes.

“The *in vivo* study lacks a control of liposomal 9AC. The tumor is very slow growing even after inoculation of 10 million cells. The reason is not clear.”

We previously answered this concern by mentioning that we chose to compare 9AC-G^W liposomes *in-vivo* with another stable long circulating liposome, and we selected doxorubicin liposomes for that purpose since they are one of the best characterized anti-cancer liposomal drug. As mentioned, 9AC liposomes did not show any stability *in-vitro* with near 90% of drug loss within just hours. That makes it behaves probably in a similar way as “free 9AC” which was included in our study.

Concerning doxorubicin liposomes, as demonstrated in supplementary figure 11, they were not able to reach a good therapeutic efficacy in mice without causing toxicity. Advantageously, this result contrasts with liposomal 9AC-G^W.

We agree with the lack of details added to the manuscript concerning that point, and hope to make it clearer this time:

Page 22, line 466: We chose doxorubicin liposomes as the closest control for 9AC-G^W liposomes, as both are stable long-circulating liposomal formulations of anticancer compounds. The chosen dosage was based on considerations of solubility and toxicity since we wished to test a condition representing a good compromise between therapeutic efficacy and toxicity.

We also wish to add some details about the slow growing rate of MDA-MB468 in mice xenograft models, we believe that such models are more representative of the real tumor growing rate in humans:

Page 21, line 464: MDA-MB468 tumors grow slowly in mice, which might more closely mimic the growth rate of tumors in humans, and may therefore be a more appropriate model of cancer than faster growing xenografts.

Reviewer #6

We thank reviewer #6. The comments were helpful to improve the manuscript. Please find our responses to the expressed concerns below:

1. Camptothecins have been noted to be very unstable in aqueous media due to their closed lactone ring, which opens at alkaline pH (pH>5.5) forming inactive open lactone form. The rates of hydrolysis of CPT in a PBS (pH 7.4) have been studied and it has been shown that only 10% activity remained after 2 hours in PBS. Moreover, liposome development has been targeted to stabilize the compounds in the bilayer where its lactone is protected from the hydrolytic aqueous phase. Keeping these in mind and use of pH 8.5 inside liposomes, I think authors should describe what actually they are measuring in the liposomes i.e. is it the of ACG closed lactone or ACG open lactone? The explanation makes it confusing, as it could be a mixture of components inside such as 9AC-GW open lactone, 9AC-GW closed lactone, calcium salts of lactone, calcium salt of glucuronate etc.). This in turns requires explanation for different precipitation phenomenon inside the liposomes.

Thank you for this comment. It is true that 9AC-G^W is present with an open lactone ring due to the high pH inside liposomes as drawn in figure 1. This is actually an advantage since the open lactone ring may further enhance the retention of drug inside liposomes beyond what is provided by the glycosidic switch. As mentioned by reviewer #6, the opened lactone form of camptothecins display poor anticancer activity. However, in our case, the glycosidic switch is enzymatically removed after liposomes reach lysosomes in target cells (Fig 1, Fig 5 & supplementary figure 7). The lactone ring in the drug can spontaneously reform due to the low pH in lysosomes (pH ~4.5). Thus, active drug is expected to be generated directly inside target cells, which may allow very effective inhibition of topoisomerase I.

We added the following summary to the revised manuscript:

Page 14, line 289: It is worth noting that camptothecins drugs (such as 9AC and BQC) might be well suited for improved retention due to the presence of a lactone ring. Indeed, the lactone ring will be found under an open form at the high pH inside the liposomes, revealing an additional charged carboxylate group for enhanced retention.

2. Authors showed that the switch molecules are hydrolyzed to water soluble counterparts (Fig 2 Conversion of 9AC-GL to 9AC-GW), however,. they never emphasized on the fact that it is not actually the conversion of 9AC-GL to 9AC-GW and rather it is conversion to open lactone 9AC-GW. Authors need to bring this to notice in the manuscript and also explain the disposition occurring in the cancer cells under acidic conditions, which is essential for activity of 9AC (about lactone ring).

Thanks you for the suggestion to improve the clarity of our manuscript concerning the lactone-carboxy forms of the CPT drugs. In our manuscript, the term “9AC-G^W” refers to both the closed or opened lactone forms. The open lactone is the form present inside liposomes. After being released in the lysosomes, the closed lactone form can spontaneously form due to the acidic pH conditions in lysosomes for escape into the cytosol and for improved potency.

Additional information was added to the manuscript:

Page 8, line 159: For all analysis, camptothecin drugs were incubated at pH 2.9 before injection to allow the reformation of the lactone ring.

Page 13, line 258: We wish to emphasize that the term “-G^W” used with camptothecin drugs refers either to the closed lactone or the opened carboxyl forms, which depends on the environmental pH. As an example, “9AC-G^W” inside the liposomes, refers to the open carboxyl form, however, when released inside the lysosomes “9AC-G^W” is expected to reform the closed lactone ring form.

Page 14, line 292: After lysosomal routing and enzymatic degradation of the liposomes⁴, the closed lactone form can spontaneously reform at the acidic pH in lysosomes for rapid escape into the cytosol and enhanced anticancer activity afforded by the lactone form of camptothecins⁵.

3. This in turn also warrants the explanation for further disposition and activity of other molecules (glucuronic acid, calcium and benzyl alcohol) generated at equimolar or higher concentrations as drug after the glucuronidase cleavage. Don't these molecules impart their effects on the cytotoxicity or other effects?

CPT drugs are orders of magnitude more potent than the other mentioned components of the switch drugs. To our knowledge, no toxicity of **glucuronic acid** has been observed or reported. One study specifically focused on the preclinical toxicological study of D-glucuronic acid⁶ and concluded that: “A one-month treatment of rats (at a single daily dose of 50, 250, and 500 mg/kg, i.p.) and dogs (50 mg/kg, i.v.) induced neither functional nor morphological changes in hemopoietic and lymphoid organs, kidney, heart, as well as in the digestive, nervous, hemostatic, and fibrinolytic systems.”. From their results, even very high doses of D-glucuronic acid only produced local irritation after intraperitoneal injection.

High **calcium** in blood results in a pathology called hypercalcaemia when levels are higher than 2.6 mmol per liters, a level which is expected to be higher than the dose received at therapeutic efficacy of the liposomes.

Benzyl alcohol was reported to have low acute toxicity with LD50 dose greater than 1 g per kg in animals. It also has been widely used as an antimicrobial preservative in medication at 0.9% for infants, but since 1982, the FDA issued recommendations to warn pediatricians against the use of benzyl alcohol containing fluids and diluents intended to be used in newborn infants whenever possible. This concern is due to the immature metabolic and excretory pathways in infants, particularly in low birth weight infants. However, at therapeutically effective dosage of drug-loaded liposomes, the amount of benzyl alcohol produced after beta-glucuronidase activation is estimated to be lower than its toxic dose, especially in adults. We made a simple estimation based on the therapeutic dosage (10 mg/kg): If 100% of 1% (targeting successfully the tumor) of the total liposomes injected were processed, it would produce only 2.9 nmol of benzyl alcohol which is equal to 15.6 µg/kg, thus almost 65000 times below the LD50.

Altogether we think that compared to the toxicity of the parental compounds (and especially camptothecins) the byproducts generated after enzymatic cleavage will only have a minor anti-proliferative effect.

We added the following to the manuscript:

Page 20, line 430: Several byproducts of the degradation of the glycosidic switch, including benzyl alcohol and glucuronate, are produced during liposomal processing within the cells. However, their toxicity is several orders of magnitude below the toxicity of camptothecins.

4. Secondly, when authors have used methanol for esterification, this makes it important that authors explain the release of methanol upon hydrolysis. As methanol could be toxic to the cells, they need to rationally justify based on the quantity of the methanol that the cells will be exposed upon dosing and give relevant deduction.

Methanol is first used for the synthesis of the lipid soluble forms (G^L forms) and is removed after HPLC purification by rotary evaporation. Traces of methanol are generated inside liposomes during internal conversion of the G^L forms to G^W forms at equimolar quantities. Methanol generated inside the liposomes can diffuse through lipid bilayers⁷ and is diluted in the external buffer of the liposomes. Later, the methanol is removed in a similar way to the non-encapsulated drugs by gel filtration on a G50 size exclusion column. For this reason, the presence of methanol is unlikely when exposing the cells to loaded liposomes. If 100% of the methanol produced in the liposomes was retained, the dose of methanol that would be injected would be equal to 290 nmol for a 20 gr mouse, or only 9.3 μ g (equal to 11.8 nL), at therapeutic effective dosage (10mg/kg).

We added the following to the manuscript:

Page 15, line 315: After loading, the glycosidic switch is exposed to an internally high pH and undergo saponification, releasing methanol. As methanol diffuses through lipid bilayers⁷, it is not retained internally and is diluted in a larger external volume. Removal of non-encapsulated compounds after loading also contributes to remove the methanol generated during that step.

5. Authors' generalization to apply the switch methodology to a wide range of compounds is highly speculative. Even though 9AC, BQC and 4MU had similar log P values, their switch versions i.e. W and L versions gave different partitioning to the molecules with logP of 0 for 4MU-GL which very low as compared to logP of 9AC-GL and BQC-GL. This important factor needs to be considered to make generalization. Moreover, the application is only possible due to the lactone ring of the aminocamptothecin, which is an important factor to consider for the switch-based modification of molecule otherwise it is just another molecule as doxorubicin. This in turn makes the choice of 4MU improper.

It is likely that the presence of the lactone ring on camptothecin compounds helps for the retention of the drugs inside the liposomes. We observed as well that 4MU- G^W could not be found at similar extent inside liposomes compared to 9AC- G^W and BQC- G^W . We have linked this result with the fact that the logP of 4MU- G^L is not lipophilic enough to permit the drug loading to be as efficient as the other two compounds. We actually think that the use of 4MU-G provides honest estimation about the generalization of this method. In addition we also provided a comparison of the loading efficacy between parental 4MU and 4MU-G and

observed a significant improvement of the loading capability using the glycosidic switch, also justifying its use for other compounds than camptothecins.

Additional information can be found at:

Page 14, line 289: It is worth noting that camptothecin drugs (such as 9AC and BQC) might be well suited for improved retention due to the presence of a lactone ring. Indeed, the lactone ring will be found under an open form at the high pH inside the liposomes, revealing an additional charged carboxylate group for enhanced retention.

Page 17, line 363: A beneficial effect of the camptothecin lactone ring on improved retention can also be considered, as the lactone ring is present in the open charged carboxyl form at the high pH inside liposomes. However, the poor retention of 9AC and BQC in the same liposomes (with high internal pH) demonstrates that opening of the lactone ring by itself is insufficient to achieve good retention of the drugs inside liposomes.

Page 16, line 328: 4MU-G^L was loaded less efficiently than 9AC-G^L and BQC-G^L. We suspect that the partitioning of the modified drugs between the aqueous and the organic phases is important for efficient loading. For example, 4MU-G^L is only very slightly lipophilic (Log P ~ -0.002) and it appears that a clear partitioning towards the organic phase is preferred, as observed with BQC-G^L (Log P ~ 1.3) and 9AC-G^L (Log P ~ 1.1) for efficient loading. We suspect that esterification of the glycosidic switch with longer carbon chains (i.e. ethanol, propanol, etc.) might help to increase the log P values of such compounds and therefore the loading efficiency. In the end, the loading efficiency of 4MU-G^L was still superior to the parental compound 4MU (**Fig. 3c**).

6. Authors responded to the reviewer 2's comment on the dose selection of free and liposomal formulation. Though authors tried to justify the dose difference due to limitations of solubility, it would have been better to estimate the effectiveness at the same dose level as in clinical settings, iv anticancer drugs are always given by infusion which to some extent bypasses the solubility limitation. Anyhow, the dosing is also possible in the fractions, if authors had considered that.

We are considering deeper research about in-vivo activity of glycosidic switch related drug loading. For this article, we have focused more about the proof of concept and we understand that further animal studies are necessary for future directions. As mentioned previously, we chose dosage based on the maximum tolerance dose, in order to show the maximum benefit that we could acquire from each group tested

We added the following to the manuscript:

Page 22, line 468: The chosen dosage was based on considerations of solubility and toxicity since we wished to test a condition representing a good compromise between therapeutic efficacy and toxicity.

7. Authors have given the in vivo activity of 9AC-GL liposomes against the free drug and parental drug (Fig 6). The results show that there is an unprecedented therapeutic activity of the liposomes, which doesn't even allow the tumor to grow. Further, in Supplementary fig 10 a, the difference in tumor size between the Doxisome and 9AC-GL liposomes is unbelievable (~65 vs 0; i.e. >60 times). However, this doesn't

coincide with the IC50 assay in which IC50 values are (2.4 vs 0.15, i.e. >24 times). Usually, the *in vivo* studies turn out to give results, which most probably are, lower as compared to *in vitro* and at the highest luck, same as *in vitro* results. The results need appropriate justification.

We noticed this phenomenon as well, and we believe that this is due to the potency of the drug loaded inside the liposomes. For example, in the case of doxorubicin, the potency is relatively low, and it resulted in no additional clinical benefit compared to free drug, excepted from cardioprotection effect. The enhanced permeability and retention effect, which is so far the only mechanism to target nanoparticles passively to tumors, is relatively poor in terms of efficiency (around 1% of total injected dose target tumors at most). Consequently, doxorubicin liposomes *in-vivo* would not be able to display as good efficiency as liposomes loaded with a more potent drug, such as camptothecins. For the same amount of targeted liposomes, doxorubicin might not be able to inhibit tumor cells, as they would do in an *in-vitro* assay where all the nanoparticles are exposed directly to the cells. On the other hand, even with low targeting efficiency, liposomes loaded with camptothecins would potentially be able to inhibit tumor growth in a much more effective way. This phenomenon was discussed and published recently using monomethyl auristatin conjugated nanoparticles.

Additional information can be found at:

Page 22, line 481: The *in-vivo* anticancer activity of 9AC-G^W liposomes was surprisingly more effective than Doxisome; 9AC-G^W liposomes cured tumors even though the difference in the *in-vitro* IC₅₀ values of 9AC-G^W liposomes and Doxisome was only about 20 fold (**Supplementary table 1**). This result suggests that encapsulation of highly potent drugs in liposomes may be highly beneficial to help overcome inherent limitations in tumor drug accumulation afforded by the EPR effect. We believe that in order to counterbalance the low efficacy of EPR delivery, the potency of nanomedicines is a critical factor for successful tumor therapy. The recent demonstration of the benefits of nanoparticle drug delivery of a highly potent antiproliferative compound, monomethylauristatin E (MMAE), also supports this idea¹. Since the potency of 9AC is about two orders of magnitude greater than doxorubicin, we suspect that the benefit obtained by 9AC-G^W liposomes is significantly greater than with doxorubicin liposomes, as demonstrated by our *in-vivo* results. In a similar way, early antibody-drug conjugates (ADCs) that employed unstable linkers to attach moderately potent anticancer agents such as doxorubicin or methotrexate produced disappointing clinical benefits^{8, 9}, whereas later ADCs that employed stable linkers to attach highly potent drugs have displayed spectacular clinical anticancer activity^{10,11}.

8. Release studies of the liposomes were performed in PBS. These studies need to be performed in three mediums (pH 7.4, pH 6.4 and pH 5.5) to represent the uptake of liposomes in the normal tissue, cancer interstitium and inside cancer cells. Reported release study is sufficient to justify the retention of the drugs in liposomes during circulation.

We understand the concern. However, we worry about misleading our audience if performing a release assay at a lower pH. Our concern is that the readers might get the wrong impression that we are trying to claim that the liposomes are made so that they can release their cargo at low pH environment (i.e. lysosomes). In reality these liposomes are very likely to be stable at

lower pH. We believe that the mechanism of cargo release inside the cells is due to the enzymatic machinery located inside the lysosomes (i.e. lipases) rather than the low pH itself.

We added the following to the manuscript:

Page 14, line 292: After lysosomal routing and enzymatic degradation of the liposomes⁴, the closed lactone form can spontaneously reform at the acidic pH in lysosomes for rapid escape into the cytosol and enhanced anticancer activity afforded by the lactone form of camptothecins⁵.

9. As authors also agree with the fact that EPR doesn't ensure total delivery of the nanocarrier to tumors, yet it is the only mechanism, which is primarily important for preferential accumulation of nanoparticles in tumors as opposed to other normal tissues due to the leaky vasculature and inefficient lymphatic drainage in tumors. However, distribution to other normal organs is what cannot be avoided completely. Hence, given the very low IC₅₀s of the compounds, authors should consider a few in vitro studies to evaluate the cytotoxicity to normal cells i.e. liver cells, normal tissue cells (representative of the site of the cancer).

Thank you for raising this important question. We measured the in-vitro anti-proliferation activity of 9AC-G^W loaded liposomes with non-cancerous human fibroblasts (GM637), BALB/c normal liver cells (BNL CL.2) and mouse fibroblasts (3T3). We noted that the IC₅₀ value toward human fibroblasts was higher than any of the cancerous cells tested (1.6 μM vs. 0.1 ~ 1.2 μM) while murine normal fibroblast and liver cells IC₅₀ values were even greater, 16 and 22 μM, respectively. We believe that 9AC-G^W liposomes demonstrate less toxicity to normal cells due to 2 main reasons: 1. The doubling time of normal cells is usually slower than cancerous cells, and 2. Cancerous cells overexpress lysosomal enzymes such as beta-glucuronidase.

Additional information can be found at:

Page 20, line 422: We also examined the anti-proliferative effect of 9AC-G^W liposomes to "normal" human fibroblasts (GM637), murine liver cells (BNL CL.2) and murine fibroblasts (3T3) (**Supplementary fig. 9**). We noted that the IC₅₀ values of human fibroblasts treated with 9AC-G^W liposomes was higher than any of the human cancer cells tested (1.6 μM vs. 0.1 ~ 1.2 μM) while murine fibroblasts and liver cells were even less sensitive to 9AC-G^W liposomes with IC₅₀ values of 16 and 22 μM, respectively. We believe that 9AC-G^W liposomes demonstrate reduced toxicity to normal cells due to two main reasons: 1) The doubling time of normal cells is usually slower than cancerous cells, and 2) The overexpression of lysosomal enzymes such as beta-glucuronidase by cancerous cells¹².

Supplementary Figure 9: Antiproliferative activity of 9AC-G^W liposomes with non-cancerous cells. Human fibroblast cells (GM637), murine liver cells (BNL CL.2) and murine fibroblasts (3T3) were seeded overnight at 5000 cells per well in 96 well-plates. Serial dilutions of 9AC-G^W loaded liposomes were added to the cells for 24 hours. Cell proliferation was evaluated by total ³H-thymidine incorporation. Error bar: SD, *n* = 3.

1. Qi R, *et al.* Nanoparticle conjugates of a highly potent toxin enhance safety and circumvent platinum resistance in ovarian cancer. *Nat Commun* **8**, 2166 (2017).
2. Zhao YM, *et al.* Augmenting drug-carrier compatibility improves tumour nanotherapy efficacy. *Nature Communications* **7**, (2016).
3. Johnston MJ, *et al.* Therapeutically optimized rates of drug release can be achieved by varying the drug-to-lipid ratio in liposomal vincristine formulations. *Biochim Biophys Acta* **1758**, 55-64 (2006).
4. Dijkstra J, van Galen M, Regts D, Scherphof G. Uptake and processing of liposomal phospholipids by Kupffer cells in vitro. *Eur J Biochem* **148**, 391-397 (1985).
5. Redinbo MR, Stewart L, Kuhn P, Champoux JJ, Hol WG. Crystal structures of human topoisomerase I in covalent and noncovalent complexes with DNA. *Science* **279**, 1504-1513 (1998).
6. Karpova GV, *et al.* [Preclinical toxicological study of D-glucuronic acid]. *Eksp Klin Farmakol* **64**, 68-70 (2001).
7. Patra M, *et al.* Under the influence of alcohol: the effect of ethanol and methanol on lipid bilayers. *Biophys J* **90**, 1121-1135 (2006).
8. Meyer DL, Senter PD. Recent advances in antibody drug conjugates for cancer therapy. *Annual Reports in Medicinal Chemistry, Vol 38* **38**, 229-237 (2003).
9. Sievers EL, Senter PD. Antibody-Drug Conjugates in Cancer Therapy. *Annual Review of Medicine, Vol 64* **64**, 15-29 (2013).
10. Doronina SO, *et al.* Development of potent monoclonal antibody auristatin conjugates for cancer therapy. *Nature Biotechnology* **21**, 778-784 (2003).
11. Wu AM, Senter PD. Arming antibodies: prospects and challenges for immunoconjugates. *Nature Biotechnology* **23**, 1137-1146 (2005).
12. Rubin DM, Rubin EJ. A minimal toxicity approach to cancer therapy: possible role of beta-glucuronidase. *Med Hypotheses* **6**, 85-92 (1980).

Reviewers' Comments:

Reviewer #5:

Remarks to the Author:

The authors have now thoroughly revised their manuscript in which they describe the reversible attachment of a glycosidic switch to drug molecules allowing their alteration between a lipid-soluble and a water-soluble state.

With this revision the points raised by the individual reviewers have now been sufficiently addressed. I support the publication of the manuscript after minor editorial changes.

Reviewer #6:

Remarks to the Author:

The authors have addressed all the comments and the manuscript is improved from before